# Fetal and postnatal metal dysregulation in autism

Manish Arora[1], Abraham Reichenberg[1,2], Charlotte Willfors[3,4], Christine Austin[1], Chris Gennings[1], Steve Berggren[3,4], Paul Lichtenstein[5], Henrik Anckarsäter[6], Kristiina Tammimies[3,4] & Sven Bölte[3,4]

Genetic and environmental factors contribute to the etiologies of autism spectrum disorder (ASD), but evidence of specific environmental exposures and susceptibility windows is limited. Here we study monozygotic and dizygotic twins discordant for ASD to test whether fetal and postnatal metal dysregulation increases ASD risk. Using validated tooth-matrix biomarkers, we estimate pre- and post-natal exposure profiles of essential and toxic elements. Significant divergences are apparent in metal uptake between ASD cases and their control siblings, but only during discrete developmental periods. Cases have reduced uptake of essential elements manganese and zinc, and higher uptake of the neurotoxin lead. Manganese and lead are also correlated with ASD severity and autistic traits. Our study suggests that metal toxicant uptake and essential element deficiency during specific developmental windows increases ASD risk and severity, supporting the hypothesis of systemic elemental dysregulation in ASD. Independent replication in population-based studies is needed to extend these findings.

[1] Department of Environmental Medicine and Public Health, Icahn School of Medicine at Mount Sinai, One Gustave L Levy Place, Box 1057, New York, New York 10029, USA. [2] Seaver Autism Center for Research and Treatment, Department of Psychiatry, Icahn School of Medicine at Mount Sinai, One Gustave L Levy Place, Box 1230, New York, New York 10029, USA. [3] Department of Women's and Children's Health, Center of Neurodevelopmental Disorders (KIND), Karolinska Institutet, Floor 8, Gävlegatan 22, SE-11330 Stockholm, Sweden. [4] Center for Psychiatry Research, Stockholm County Council, Norra Stationsgatan 69, Floor 7, SE-11364 Stockholm, Sweden. [5] Department of Medical Epidemiology and Biostatistics, Karolinska Institutet, SE-171 77 Stockholm, Sweden. [6] Institute of Neuroscience and Physiology, University of Gothenburg, Box 430, SE-405 30 Göteborg, Sweden. Correspondence and requests for materials should be addressed to M.A. (email: manish.arora@mssm.edu) or to S.B. (email: sven.bolte@ki.se).

Autism spectrum disorder (ASD) affects 1–2% of all children born in Europe, North America and other developed regions, and is defined by persistent alterations in social communication and interaction alongside restricted, repetitive patterns of behaviour, interests or activities[1]. ASD causes significant impairment in social, occupational or other important areas of functioning[1,2]. Psychiatric comorbidity is common, including attention deficit hyperactivity disorder (ADHD), affecting up to 30% of children with ASD[3]. Heritable factors account for at least 50% of ASD risk with the remainder attributable to environmental factors acting alone or through interaction with genetics[4,5]. While substantial progress has been made in identifying genetic determinants of autism risk[6] environmental factors, especially modifiable environmental exposures, remain understudied.

Fetal and early childhood exposure to toxic metals and deficiencies of nutritional elements have been linked with several adverse developmental outcomes frequently associated with ASD, including intellectual disability, and language, attentional and behavioural problems[7]. Animal studies have demonstrated that the effects of various metals on brain development could be mediated through dysregulation in neurotransmission, and alterations in frontal and subcortical brain structures[8], several of which have also been implicated in ASD[9]. Therefore, environmental and dietary exposure to metals are potentially important etiological factors in ASD[9].

Studies of toxic metals and nutritional elements in ASD have yielded mixed results. Although several studies reported higher concentrations of toxic metals in ASD, including the neurotoxin lead, others found no association[10]. However, previous research shows methodological shortcomings that limit interpretation and generalizability of the findings. First, elemental exposure has frequently been estimated using concentrations in blood or other biomarkers post-diagnosis[11]. Second, most previous studies recruited patient series rather than enroling participants from the population. Third, assessing the contribution of environmental factors in the etiology of ASD needs control for genetic factors. Investigating family members, especially contrasting discordant monozygotic (MZ) twins, is a powerful strategy for uncovering disease-associated environmental factors independent of underlying genomic sequence variation[12].

Given these methodological restraints and the high likelihood of metal exposure in the environment[7], rigorous research is required that provides a more accurate assessment of the role of metals in the etiology of ASD. To that goal, we developed tooth-matrix biomarkers that directly measure fetal and postnatal exposure to multiple metals, and recruited twins from population-based cohorts from whom naturally shed deciduous teeth could be collected. We enroled 26.3% of the base study population of the Roots of Autism and ADHD Twin Study in Sweden (RATSS) through a non-random selection process (Fig. 1). RATSS applies deep phenotyping on a significant subset (currently 11.3%) of all ASD discordant twins in Sweden in the specified age range[13]. Importantly, since the RATSS only recruits participants older than 8 years, and children only shed teeth until the age of 12 years, the study we report here represents 50% of the RATSS base population who are of tooth-shedding age. In addition to our primary study in RATSS, we were able to obtain teeth from an ASD-discordant DZ twin pair in the United States, which were analysed separately (Supplementary Material).

The teeth samples we collected from the twins were analysed using recently developed and rigorously validated tooth-matrix biomarkers, which objectively reconstruct fetal and early post-natal exposure to multiple metals with detailed temporal resolution[14–19]. We have developed these direct fetal biomarkers that can be collected at the time of diagnosis in childhood and provide time-series data from the second trimester onwards on multiple metal exposures (overview of these biomarkers shown in Fig. 2)[14]. We tested the hypothesis that prenatal and early life exposure to metal toxicants or deficiency of essential elements during critical developmental windows are associated with ASD. Our primary focus was on lead, an established neurotoxicant that has been implicated in ASD[20], and manganese, an essential nutrient with emerging evidence of neurotoxicity[21]. Both these elements have been associated with attentional and behavioural outcomes relevant to ASD[7,21]. We also examined zinc, an essential element that is critical for the maintenance of health and central to the regulation of many metal transport mechanisms[22]. Moreover, we undertook exploratory analysis of seven other elements.

## Results

**Participant characteristics.** We recruited twins from nation-wide twin registries and by advertisements in Sweden (see Methods section and Fig. 1)[23,24]. Characteristics of the study sample are presented in Table 1. A total of 154 twin pairs had participated in RATSS as of September 2016, which includes 11.3% of all ASD discordant twins in Sweden in the specified age range[13]. From those participants, this study included 32 complete twin pairs and 12 individuals from twin pairs whose sibling did not donate a tooth. Out of the 32 complete twin pairs, 17 were MZ pairs and 15 DZ pairs including two opposite-sex pairs.

**Metal distributions in typically developing and ASD twins.** Our tooth-matrix biomarkers provide time-series data on uptake of multiple elements from the second trimester to early childhood. Figure 3 shows typical elemental distributions we observed in teeth of ASD cases and controls. We first examined distributions of elements in non-ASD MZ twins to establish characteristic metal uptake patterns at different developmental times. Figure 3a shows close concordance for lead, manganese and zinc in non-ASD control MZ twins (other elements show a similar pattern, Supplementary Fig. 3). The observed metal distributions were in agreement with previous studies using tooth-matrix biomarkers[14,19]. Specifically, manganese levels declined rapidly over the prenatal period to birth and continued to decrease at a slower rate postnatally. Zinc levels were steady prenatally in our sample teeth with a marked decrease around birth. There is no typical distribution of lead in teeth related to developmental age.

Examining distribution of elements between MZ twins discordant for ASD revealed a non-characteristic pattern of element-specific differences (Fig. 3b shows typical examples). Manganese levels were lower in the affected twin pre- and postnatally. A complex pattern emerged for zinc; in non-ASD co-twins there was a decline in zinc around birth but in ASD cases this drop in zinc levels occurred earlier during the prenatal period and subsequently zinc levels showed a marked increase postnatally, surpassing the levels in their non-ASD co-twins. Lead levels were generally higher in the affected twin and this difference was greatest after birth. In an ASD discordant DZ twin sample recruited from an autism clinic in the United States, we found similar differences in zinc and manganese between the ASD-affected and healthy twin (see Supplementary Fig. 2). In MZ pairs concordant for ASD (Fig. 3c), the differences in metal distribution amongst twins were smaller than those observed in discordant pairs.

**Developmental periods of metal dysregulation in ASD.** To uncover critical windows of susceptibility to metal dysregulation, we used distributed lag models (DLMs), a statistical method for

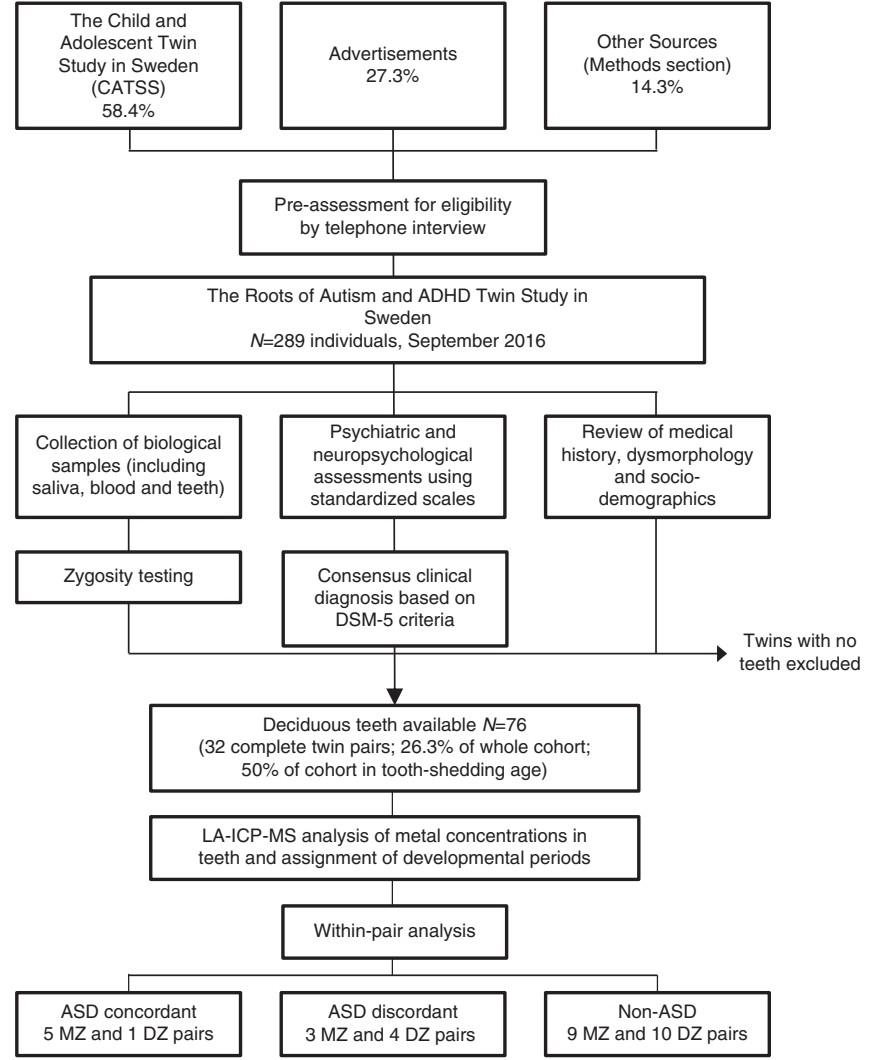

**Figure 1 | Roots of Autism and ADHD Twin Study in Sweden (RATSS).** Recruitment, case ascertainment and collection of teeth for measurement of metals.

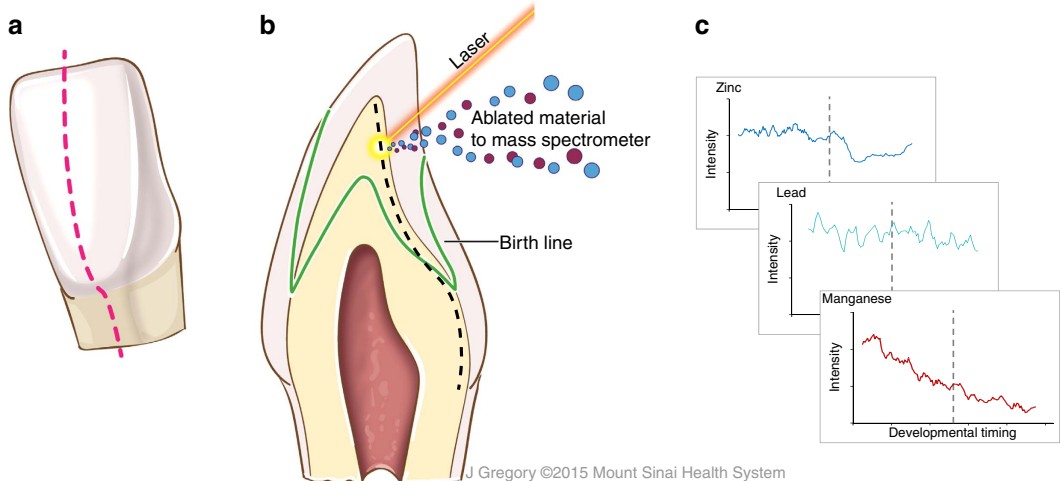

**Figure 2 | Overview of tooth-matrix biomarkers applied in this study.** (**a**) Plane in which teeth were sectioned. (**b**) Laser ablation-inductively coupled plasma mass spectrometry analysis. The surface of dentine was ablated using a laser and the ejected material transferred to a mass spectrometer. Neonatal line and other histological features are used to select sampling locations adjacent to the dentine-enamel interface extending from the dentine horn to the tooth cervix (dashed black line). (**c**) Output data where developmental timing is assigned to elemental profiles in each sample, reconstructing the history of elemental uptake over the prenatal and early childhood periods. The illustrations were created by Jill K Gregory from the Icahn School of Medicine at Mount Sinai.

**Table 1 | Characteristics of the twin study sample.**

| | ASD discordant twin pairs (N = 20 individuals)* | | ASD concordant twin pairs (N = 12 individuals) | Non-ASD twin pairs (N = 44 individuals)[†] |
|---|---|---|---|---|
| Gestational week | 34.7 ± 4.6 | | 34.7 ± 1.5 | 36.0 ± 3.5 |
| Age at assessment | 14.7 ± 3.9 | | 11.5 ± 3.0 | 13.0 ± 3.1 |
| No. females: males | 8:12 | | 4:8 | 18:26 |
| No. MZ:DZ twin pairs[‡] | 3:4 | | 5:1 | 9:10 |
| | **ASD twin** | **Co-twin** | | |
| Average birth weight in grams ( ± s.d.) | 2,116.6 ± 872.6 | 2,207.7 ± 773.1 | 2,492.7 ± 495.7 | 2,433.7 ± 761.5 |
| ADOS-2 severity score | 4.1 ± 1.8 | 2.2 ± 2.2 | 6.1 ± 2.8 | 2.0 ± 1.6 |
| SRS-2 total raw score | 74.1 ± 28.6 | 32.3 ± 25.4 | 98.7 ± 16.2 | 41.9. ± 32.3 |
| Full scale Intelligence Quotient | 93.1 ± 19.0 | 104.2 ± 11.1 | 85.4 ± 21.1 | 99.0 ± 15.3 |

ADOS-2, The Autism Diagnostic Observation Schedule Second Edition; ASD, autism spectrum disorder; DZ, dizygotic; MZ, monozygotic; SRS-2, Social Responsiveness Scale Second Edition.
*Six singletons.
†Six singletons.
‡Singletons excluded.
Data are from 32 full twin pairs and 12 singletons from twin pairs that have participated in the RATSS study.

the analysis of time-series data, particularly useful to study the effect of an exposure at a certain time point while adjusting for all the past (lagged) values of that exposure[25]. We adjusted our analysis for important covariates and also applied two levels of stringent corrections. First, we adjusted for intra-twin correlations and second, we used a Holm–Bonferroni correction to account for multiple comparisons (Methods section). In this manner, we identified specific developmental periods when metal levels varied between ASD discordant twins (7 pairs), after accounting for the difference between control twins (19 pairs; Fig. 4). We defined the critical developmental window for a metal and ASD as that region of the DLM graph where the intra-twin corrected association did not contain zero (blue bands in Fig. 4). Statistical significance is represented by Holm–Bonferroni confidence intervals (CI; vertical bars in Fig. 4).

When comparing ASD discordant twins with non-ASD control twin pairs, we found that lead levels were consistently higher in ASD cases than their non-ASD co-twins from 20 weeks before birth to 30 weeks after birth. However, after adjusting for intra-twin correlations, this association was only evident between weeks 10 to 20 postnatally. The greatest difference was observed at 15 weeks postnatally, when lead levels in ASD cases were 1.5 times higher (Holm–Bonferroni 95% CI 0.9–2.5) than in their co-twins, after taking into account average difference in control twins. Manganese levels were consistently lower in ASD cases, and this was statistically significant over two critical windows; between 10 weeks prenatally to birth and subsequently postnatal weeks 5–20. The greatest difference was observed at postnatal week 15 when cases had 2.5 times lower manganese than their co-twins (Holm–Bonferroni 95% CI 1.1–4.5 times lower). Differences of a similar magnitude were also observed 7 weeks before birth. ASD cases were also deficient in zinc from 10 weeks prenatally to ~5 weeks postnatally, with the ASD cases showing 28% lower (or 1.28 times lower) levels than their co-twins 8 weeks prenatally (Holm–Bonferroni 95% CI 22–65% lower).

Concordant pairs, where both twins have ASD, are expected to be more alike in their metal status than discordant pairs. Therefore, in a sensitivity analyses, we compared ASD discordant twin pairs with ASD concordant twins (six pairs) and observed similar patterns to our comparison of ASD discordant versus control pairs, including significantly higher lead levels in cases (Fig. 5). Furthermore, in exploratory analysis, we observed divergence between ASD cases and their co-twins in the other metals we examined. Levels of tin were higher in ASD cases and this difference was statistically significant between 20 and 16 weeks before birth (Supplementary Fig. 4). Strontium was higher

in cases throughout the period of study and this difference was most prominent during weeks 22–30 after birth. Chromium was lower in cases than their control co-twins, with differences being greatest from 20 to 15 weeks prenatally. Overall, our analyses showed broad pre- and postnatal case-control differences, for six of ten metals examined under our hypothesis-driven and exploratory analysis.

**Correlation of metals with ASD traits and clinical severity**. We examined if early life metal dysregulation had long-lasting effects on various clinical measures of ASD and autistic traits. To test this, we correlated our tooth-matrix biomarkers with the severity of ASD and autistic traits measured approximately a decade later using established clinical assessments (Autism Diagnostic Observation Schedule Second Edition (ADOS-2), and Social Responsiveness Scale Second Edition (SRS-2), respectively). Lead and manganese showed statistically significant associations with ADOS-2 or SRS-2 (Fig. 6). Manganese was inversely associated with autistic traits measured using SRS-2 (strongest association at 15 weeks, $r = -0.25$, 95% CI $-0.40$ to $-0.10$) and ASD severity on ADOS-2 (strongest association at 12 weeks, $r = -0.20$, 95% CI $-0.35$ to $-0.05$). Lead concentrations were positively associated with ADOS-2 and SRS-2 scores. The association with ADOS-2 was statistically significant from 10 weeks before birth to 30 weeks after birth, with strongest association 5 weeks prenatally ($r = 0.40$, 95% CI 0.20–0.60). Zinc and other metals analysed under exploratory analysis were not significantly associated with SRS-2 and ADOS-2.

**Discussion**
Using tooth-matrix biomarkers, which measure the uptake of multiple elements at a fine temporal resolution during early development, and a well-characterized sample of twins, we observed significant differences between ASD cases and non-ASD controls during specific pre- and postnatal periods. In ASD cases, higher lead levels were observed over the prenatal period and first 5 months postnatally. Levels of essential elements were diminished in cases at specific developmental windows. Zinc levels were lower in cases during the third trimester, while manganese levels were consistently lower in cases both pre- and postnatally, and this deficiency was highest 4 months after birth. Differences between cases and controls were also evident for multiple other elements examined in our exploratory analysis, including tin, strontium and chromium. The developmental period of maximal difference between cases and controls varied

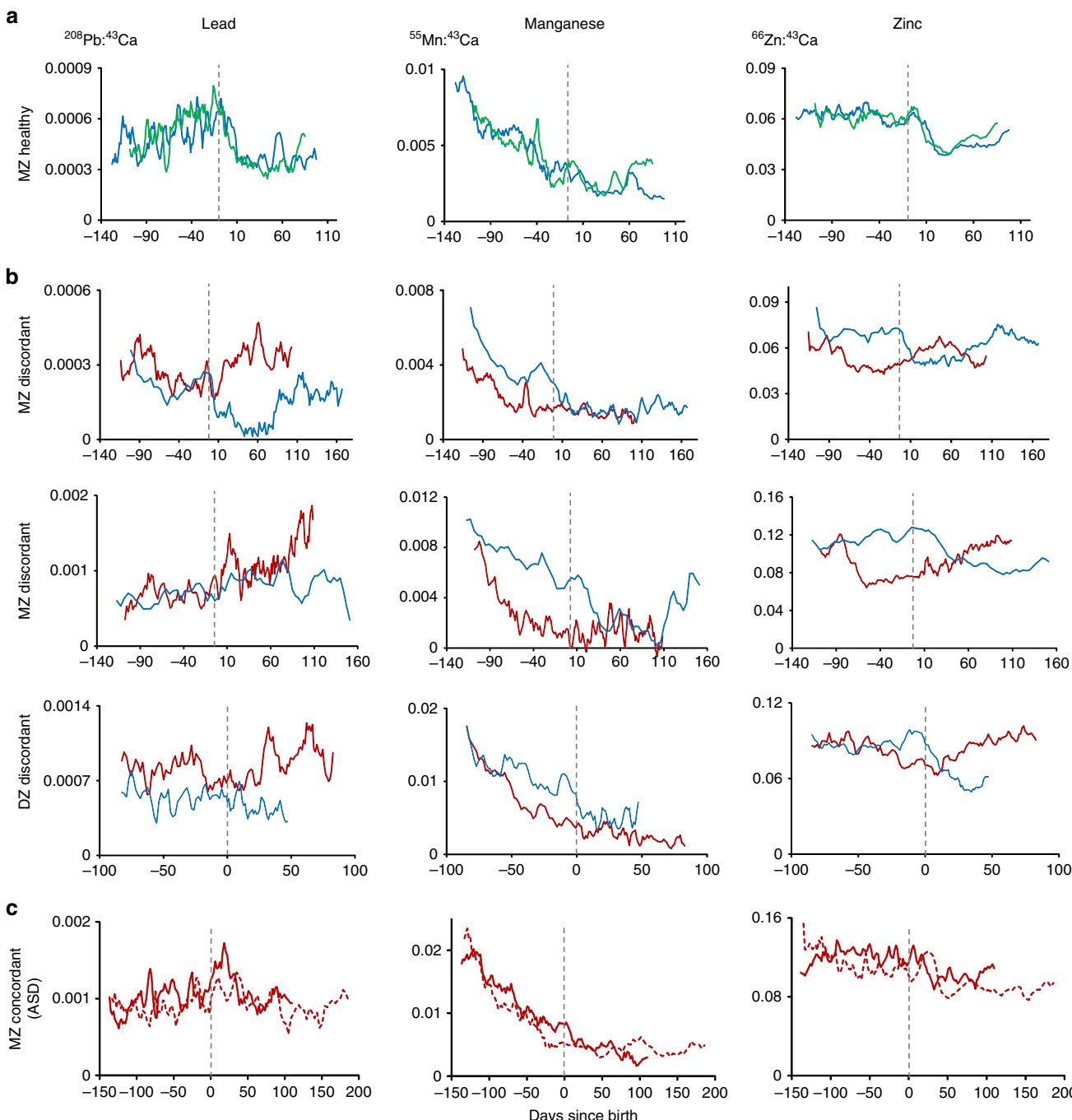

**Figure 3 | Pre- and postnatal differences in metal uptake between non-ASD control and ASD discordant and concordant twin pairs.** Patterns in lead, manganese and zinc distribution over the developmental period studied are shown for (**a**) non-ASD control monozygotic (MZ) twin pair, (**b**) ASD discordant MZ and dizygotic (DZ) twin pairs and (**c**) ASD concordant MZ twin pair. Red lines indicates ASD twins and blue/green, control co-twins. Grey dashed line denotes time of birth. Element levels are reported as metal ion counts ($^{208}$Pb, $^{55}$Mn, or $^{66}$Zn) relative to an internal standard ($^{43}$Ca).

for each element. Even a decade after the exposure, we observed moderate but statistically significant correlations between two of three metals examined under primary hypothesis and ASD severity and autistic traits across the entire sample. Taken together, this supports the hypothesis that prenatal and early childhood disruption (excess or deficiency) of multiple metals during critical developmental windows is associated with ASD, and suggests a role for elemental dysregulation in the etiology of ASD.

The influence of common genetic factors in ASD is estimated to be at least 50% (ref. 4), however currently etiologic genetic variants can be found in ∼16 to 50% of affected children,

suggesting important roles for environmental factors that have not yet been confirmed[4,6]. Among the large number of possible environmental exposures, metal toxicants and essential elements have received some attention[9]. For example, one study that compared hair metal concentrations of ASD cases with literature reference values was suggestive of reduced zinc and magnesium and elevated levels of toxicants including lead[26]. Similarly, higher blood lead levels have been reported in ASD cases[11], but in those studies it is possible that the uptake of metal toxicants or nutritional deficiencies are a consequence of autism-related behaviours and do not necessarily reflect a causal exposure

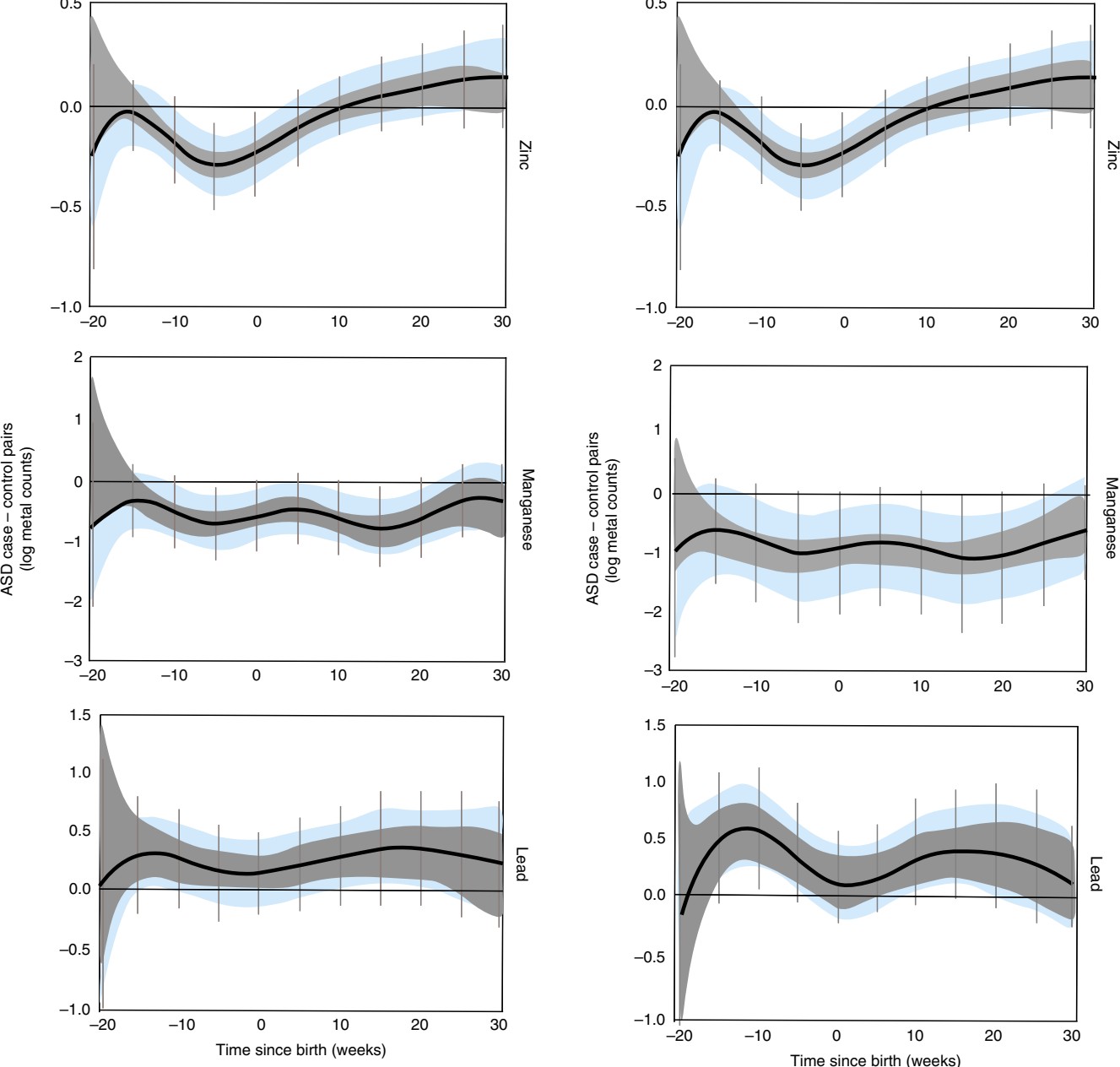

**Figure 4 | Developmental periods of maximal metal dysregulation in ASD discordant versus non-ASD control twin pairs.** DLMs for zinc, manganese and lead. Black line represents mean of paired differences in metal levels between twins discordant for ASD (3 MZ; 4 DZ) and non-ASD control pairs (9 MZ; 10 DZ). Data are shown as 95% bands unadjusted (grey band) and adjusted (blue band) for intra-twin correlations. Grey vertical lines are Holm–Bonferroni corrected family-wise 95% CIs for 11 comparisons (every fifth week). Values above zero represent increased levels in ASD cases compared to their non-ASD sibling after taking into account average difference in control twins. A critical window is defined as an area of the DLM adjusted for intra-twin correlations (blue band) that does not include zero. Models were adjusted for sex, zygosity, gestational age, the average birth weight of the twin pairs, and the s.d. of the birth weight in the twin pairs.

**Figure 5 | Developmental periods of maximal metal dysregulation in ASD discordant versus ASD concordant twin pairs.** DLMs for zinc, manganese and lead. Black line represents mean of paired differences in metal levels between twins discordant for ASD (3 MZ; 4 DZ) and ASD concordant pairs (5 MZ; 1 DZ). Data are shown as 95% bands unadjusted (grey band) and adjusted (blue band) for intra-twin correlations. Grey vertical lines are Holm–Bonferroni corrected family-wise 95% CIs for 11 comparisons (every fifth week). Values above zero represent increased levels in ASD cases compared to their non-ASD twins after taking into account average difference in concordant twins. A critical window is defined as an area of the DLM adjusted for intra-twin correlations (blue band) that does not include zero. Models were adjusted for sex, zygosity, gestational age, the average birth weight of the twin pairs, and the s.d. of the birth weight in the twin pairs.

prior to the onset of autism symptoms[27]. Previous studies have used teeth from ASD cases to measure cumulative exposure to metals, but those studies ground whole teeth not taking advantage of the incremental microstructure that provides detailed temporal

information[28]. They found no significant differences in lead, zinc or manganese between cases and controls. Our sample included MZ and DZ twins and substantially more detailed family data, and our tooth-matrix biomarkers also allowed us to investigate

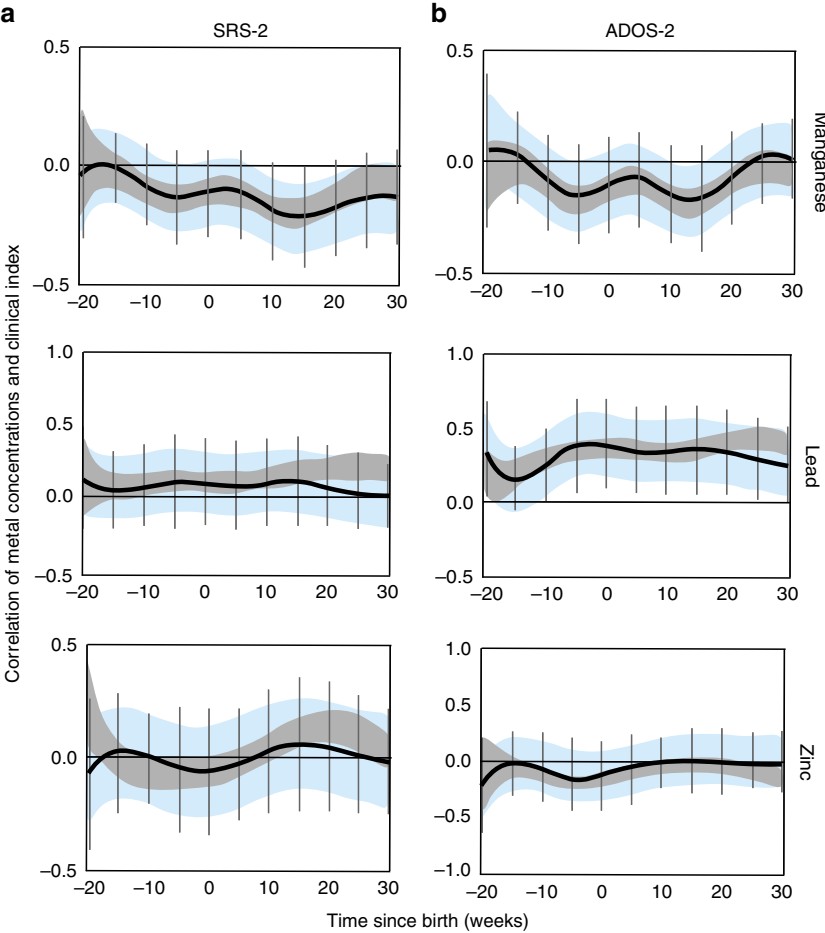

**Figure 6 | Association of elemental concentrations with measures of autism traits and clinical severity.** (**a**) Correlation of manganese, lead and zinc with Social Responsiveness Scale-2 (SRS-2) at different developmental time points. (**b**) The correlation of these elements with Autism Diagnostic Observation Schedule-2 (ADOS-2). Data are shown as 95% bands unadjusted (grey band) and adjusted (blue band) for intra-twin correlations. Grey vertical lines are Holm–Bonferroni corrected family-wise 95% CIs for 11 comparisons (every fifth week). Values above zero represent positive correlation between metal uptake and clinical index score. A critical window is defined as an area of the intra-twin adjusted correlation (blue band) that does not include zero. Models are adjusted for gender, gestational age and birth weight.

the timing of exposure during discrete developmental periods over the pre-and postnatal periods.

Early signs of ASD first manifest 6–12 months after birth[29], suggesting a narrow window for environmental factors to contribute to ASD risk. Our results support this hypothesis, as the timing of divergence between ASD cases and their co-twins in lead, zinc and manganese was evident prenatally and in the early postnatal period. These differences at specific developmental periods in early life for a range of toxic and essential elements suggest that there are multiple mechanisms leading to dysregulation, and endorse the higher vulnerability of the mid- to late-fetal period to the pathophysiological mechanisms underlying ASD.

Our findings along with other recent studies bolster the premise of joint interaction of environmental exposures with genetic variations in the etiology of ASD. Notably, many of the genes associated with ASD are also linked with elemental homeostasis[9,30], and it is intriguing that genes implicated in ASD converge to specific neuronal co-expression networks especially during the same critical early developmental periods we have observed in this study[31–33]. Furthermore, the capacity of the brain to regulate metals varies with developmental age suggesting *in utero* fluctuations in expression of metal transporter genes. Transcriptional alterations in response to chemical

exposures that mimic changes seen in brain samples from autism patients have also been reported recently[34].

A mechanistic framework of prenatal elemental dysregulation in ASD should consider the maternal–fetal interface, epigenetic modification of genes coding for metal transporters, and the central role of zinc in regulating homeostasis of multiple metals. Recent evidence from human and animal studies suggests that under stress, placental transfer of nutrients to the fetus is disrupted, and factors such as prenatal stress and placental inflammation increase hyperactivity in offspring[35,36]. In the case of MZ twins sharing a placenta, external stressors that affect only one twin may induce epigenetic modification; notably, in one study fetal zinc deficiency-induced epigenetic alterations in the gene coding for the metal transporter, metallothionein-2 (ref. 37). In this regard, it is important to consider the central role of zinc as a regulator of multiple metal homeostasis pathways from early in the transcriptional process to the optimal functioning of metal transporters[9]. Therefore, upstream factors such as placental insufficiency and epigenetic alterations of metal transporters may disrupt zinc homeostasis initiating a cascade of elemental dysregulation which, in the presence of genetic predisposition, raises the risk of ASD. Overall, it is likely that alterations in metal regulation in our ASD cases involve multiple disruptions along the complex networks that regulate elemental uptake and distribution.

**Table 2 | LA-ICP-MS operating conditions.**

| NWR-193 laser conditions | | Agilent 8800 ICP-MS conditions | |
|---|---|---|---|
| Wavelength (nm) | 193 | RF power (W) | 1,350 |
| Helium carrier flow (l min$^{-1}$) | 0.8 | Argon carrier flow (l min$^{-1}$) | 0.6 |
| Fluence (J cm$^{-2}$) | 5.0 | Plasma gas flow (l min$^{-1}$) | 15 |
| Repetition rate (Hz) | 10 | Sample depth (mm) | 4.0 |
| Spot size (µm) | 35 | Scan mode | Peak hopping |
| Scan speed (µm s$^{-1}$) | 35 | Integration time (ms) | 50–55 |

LA-ICP-MS, laser ablation-inductively coupled plasma mass spectrometry.

This study has multiple strengths, such as the inclusion of an informative twin sample recruited from population-based cohorts, a rigorous diagnostic assessment, and the use of direct fetal biomarkers. Limitations of the study are a relatively small non-random sample, although the sample size was adequate to uncover significant associations after stringent statistical adjustments, and our twin sample represented a significant subsample (11.3%) of the total population of twins discordant for ASD in Sweden in the examined age range[13]. In addition, the inclusion of twins offsets the potential lower power as it allows rigorous control for underlying genetic variation. Nevertheless, caution should be exercised when generalizing our findings, and additional studies are needed in different populations, particularly larger non-twin ASD samples to corroborate our findings, and differentiate genetic and non-genetic contributions in understanding the relation between metals and ASD. Our biomarkers, while providing a direct measure of fetal exposure, do not measure exposure during the first trimester. Furthermore, while the differences we observe between cases and controls precede the onset of ASD symptoms, these data do not establish causality. Our tooth-biomarkers measure uptake of metals from all sources but do not distinguish specific routes of exposure such as diet. Our study also highlights a need to study the kinetics of metal mixtures during fetal and early postnatal development as we noted substantial shifts in metal levels between the pre-and post-natal periods. Most studies on the clearance and half-life of metals in humans are undertaken on adults and not newborns. Heterogeneous distribution of metals in enamel and dentine can occur in response to variable environmental exposures, changes in diet and due to age-dependent metabolic changes, and has been observed in humans and other mammals[38].

In conclusion, using novel biomarkers of early life exposure, we observed differences in uptake of multiple toxic and essential elements over the second and third trimesters and early postnatal periods in MZ and DZ twins discordant for ASD. The critical developmental windows for the observed discrepancies varied for each element, suggesting that systemic elemental dysregulation may serve an important role in ASD etiology.

## Methods

**Study design and participants.** Participating twins in this study are part of RATSS recruited between 2011 and 2016 (ref. 23). The study was approved by the Swedish National Ethical Review Board and all participants gave written informed consent. Potential twin participants for the RATSS are identified through nation-wide registries, including the Child and Adolescent Twin Study in Sweden[24], a population-based study of all twins born in Sweden since 1992 in which all twins are screened at age nine using the Autism, Tics, ADHD and other Comorbidities Inventory (A-TAC). Participants are identified through linking the Swedish Twin Registry to other National registries such as the Swedish National Patient Register, and regional clinical registers in Stockholm County (Child and Adolescent Psychiatry ['Pastill'], Habilitation & Health Centers) that include ICD-10 diagnostic information[39–41]. Finally, potential participants are also identified through national Swedish societies for neurodevelopmental disorders (NDDs) as well as advertisements and summons in the media. Even though the recruitment is done through different routes, >80% of the twins in RATSS are present in the Swedish twin registries.

Twin pairs were recruited into the RATSS either based on discordance for ASD (>2 points differences on the A-TAC autism subscale equalling ~1 s.d.); concordance for ASD (both twins reaching cut-off on the A-TAC autism scale); or concordance for no NDD (both twins under cut-offs for all NDD subscale on the A-TAC). For other sources of recruitment, the twins are invited if at least one twin has an ICD-10 diagnosis of autism (F84.0), Asperger syndrome (F84.5) or atypical autism/pervasive developmental disorder not otherwise specified (PDD-NOS) (F84.1, F84.8, F84.9), or a Diagnostic and Statistical Manual of Mental Disorders Fifth Edition (DSM-5) diagnosis of ASD (either parent- or registry reported). All potential participants undergo a telephone interview by a research nurse checking eligibility before invitation for assessment in RATSS. Participants in our study had additional diagnosis of NDD. In ASD discordant twins, six (86%) ASD twins and one (13%) co-twin had a diagnosis of other NDD. Ten ASD concordant twins, but only 8 (31%) individuals among the non-ASD twins, had a NDD.

Zygosity was determined by genotyping of saliva or whole-blood-derived DNA using standard methods. The genotyping was done using Infinium Human-CoreExome chip (Illumina Inc., USA). The estimating identity by descent was analysed using the PLINK software (v1.07)[42] after quality control and removal of SNPs with minor allele frequency <0.05 within the samples. All pairs of DNA samples showing $\hat{\pi} \geq 0.99$ were considered as monozygotic pairs. For few pairs a short tandem repeat kit (Promega Powerplex 21) was used to determine the zygosity.

Currently, RATSS includes 11.3% of all ASD discordant twins in Sweden in the specified age range[13]. Importantly, since the RATSS study only recruits participants older than 8 years, and children only shed teeth until the age of 12 years, the study we report here represents 50% of the RATSS base population in the age range of 8–12 years. Finally, we recruited one ASD discordant twin pair from a clinic in the United States whose data was analysed separately (Supplementary Fig. 2).

**Clinical procedures.** Medical history and sociodemographic information of the families were collected. ASD was diagnosed according to DSM-5 criteria based on clinical experts consensus and corroborated by results from the Autism Diagnostic Interview—Revised[43] and the ADOS-2 (ref. 44). Clinical severity of ASD symptoms was determined by ADOS comparison scores, and autistic traits were measured by parent reported Social Responsiveness Scale-2 (SRS-2)[45] total raw scores. General cognitive ability was assessed using the Wechsler Intelligence Scales for Children or Adults (Fourth Editions) or the Leiter Scales and the Peabody Picture Vocabulary Test (Third Edition) in cases of low verbal abilities[46–48].

**Collection and analysis of biological samples.** Parents/guardians collected the naturally shed deciduous teeth at home. Teeth were brought to the study team in person and stored at room temperature. Metal deposits in teeth are stable at room temperature, and we have validated tooth-metal biomarkers in prospective pregnancy cohorts by comparing metal concentrations in teeth with other biological and environmental matrices[14–19]. For example, earlier we validated prenatal lead levels in teeth with maternal blood lead measured during the second and third trimesters of pregnancy, and birth and childhood measures of tooth lead with lead concentrations in umbilical cord blood and serial childhood blood measures, respectively[15]. We have similarly validated metal levels in teeth against environmental samples (house dust concentrations and distance to exposure source, for example)[16,17], and also undertaken detailed animal studies with controlled exposures[18,19].

Our approach to measuring metals in teeth using laser ablation-inductively coupled plasma mass spectrometry (LA-ICP-MS) and assigning developmental times has been detailed elsewhere[14,19]. Herein, teeth are sectioned and the neonatal line (a histological feature formed in enamel and dentine at the time of birth) and incremental markings are used to assign temporal information to sampling points (Fig. 2). A New Wave Research NWR-193 (ESI, USA) laser ablation unit equipped with a 193 nm ArF excimer laser was connected to an Agilent Technologies 8800 triple-quad ICP-MS (Agilent Technologies, USA). Helium was used as carrier gas from the laser ablation cell and mixed with argon via Y-piece before introduction to the ICP-MS. The system was tuned daily using NIST SRM 612 (trace elements in glass) to monitor sensitivity (maximum analyte ion counts), oxide formation ($^{232}Th^{16}O^+/^{232}Th^+$, <0.3%) and fractionation

($^{232}$Th$^+$/$^{238}$U$^+$, 100 ± 5%). The laser was scanned in dentine parallel to the dentine-enamel junction (DEJ) from the dentine horn tip towards the tooth cervix. A pre-ablation scan was run to remove any surface contamination. Data were analysed as metal to calcium ratios (for example, $^{208}$Pb:$^{43}$Ca) to control for any variations in mineral content within a tooth and between samples. On average, each tooth was sampled at 152 locations. LA-ICP-MS operating parameters are given in Table 2.

**Statistical analysis.** We used a variation of DLMs as our primary statistical approach[25]. A DLM is a regression-based approach commonly used in the behavioural sciences for the analysis of time-series data, particularly to study the effect of an exposure at a certain time point while adjusting for all the past (lagged) values of that exposure. Using DLMs, we were able to estimate the differences in metal uptake between ASD cases and control twin pairs at discrete developmental stages, while accounting for exposures at other time points. Specifically, we applied DLMs to the detailed temporal data generated by the tooth-matrix biomarkers to detect critical developmental windows between 20–30 weeks after birth for lead, manganese and zinc (and seven other metals under exploratory analysis). We also used DLMs to identify developmental time points when metal biomarkers were significantly correlated with ASD severity and autistic traits.

We undertook three sets of analyses. The primary analysis determined developmental periods when there are disparities in elemental concentrations associated with ASD in twin pairs discordant for ASD (7 pairs) and control (non-ASD) twin pairs (19 pairs). The parameter estimate in this analysis was the smoothed mean differences in log concentrations of discordant pairs minus mean differences in control pairs, that is, $(X_{case} − X_{control}) − (X_{control-1} − X_{control-2})$. Thus, for example, when ASD cases from the discordant pairs have higher concentrations than their non-ASD co-twins, and these differences exceed the differences observed in control twin pairs, the association (as measured by the beta coefficient) is positive. In a sensitivity analysis we similarly compared ASD discordant pairs (7 pairs) with ASD concordant pairs (6 pairs). Although the sample for this analysis was smaller it provides important support for the results of the primary analysis as the discordant versus concordant twin comparison provides additional control for confounders. Finally, we examined correlations between element concentrations and ADOS-2 and SRS-2 scores using data from all participants (ASD-discordant, ASD concordant and control twins and individuals whose twin did not participate ($n = 76$)). All models were adjusted for covariates including sex, zygosity, gestational age, the average birth weight of the twin pairs and the s.d. of the birth weight within the twin pair. We used average birth weight and s.d. in our models as our unit of analysis is a twin pair, and we wanted to adjust for differences within a twin pair since even monozygotic twins can have different birth weights. For the parameter estimates, associated time-varying 95% CIs were calculated, corresponding to a statistical test on the two-sided 5% level of significance. We applied two additional corrections; first, we adjusted for intra-twin correlations through a random effect term per twin pair and, second, we used a Holm–Bonferroni correction to account for multiple comparisons. Additional details of the statistical methods are in Supplementary Methods and Supplementary Fig. 1.

**Data availability.** Data sets generated and analysed during the current study are not publically available because they contain private patient health information, but are available from the corresponding authors on reasonable request and subject to necessary clearances.

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

## Acknowledgements

We thank all the twins and their families for participating in this study. Genotyping was performed by the SNP&SEQ Technology Platform in Uppsala (www.genotyping.se). The facility is part of the National Genomics Infrastructure (NGI) Sweden and Science for Life Laboratory. The SNP&SEQ Platform is also supported by the Swedish Research Council and the Knut and Alice Wallenberg Foundation. We thank Ms. Jill Gregory, Academic Medical Illustrator at the Icahn School of Medicine at Mount Sinai, for help with preparation of figures. Support was provided by the Innovative Medicines Initiatives Joint Undertaking (grant agreement number 115300), which comprises financial contribution from the European Union's Seventh Framework Programme (FP7/2007–2013) and in-kind contributions from companies belonging to the European Federation of Pharmaceutical Industries and Associations; the Swedish Research Council (523-2009-7054; 521-2013-2531); the Swedish Research Council, in partnership with the Swedish Research Council for Health, Working Life and Welfare, Formas and VINNOVA (cross-disciplinary research programme concerning children's and young people's mental health, 259-2012-24), Stockholm County Council (20100096, 20110602, 20120067, 20140134), Stiftelsen Frimurare Barnhuset, Sunnerdahls, Handikappfond, Hjärnfonden. MA was supported by the National Institutes of Environmental Health Sciences research grants (DP2ES025453, R00ES019597, P30ES023515) and the Mindich Child Health and Development Institute pilot grants. K.T. is financially supported by the Swedish Foundation for Strategic Research and Jeanssons foundations. A.R. is supported by grants from the National Institutes of Health; grant HD073978 from the Eunice Kennedy Shriver National Institute of Child Health and Human Development, National Institute of Environmental Health Sciences, and National Institute of Neurological Disorders and Stroke; grant MH097849 from the National Institute of Mental Health; and by the Beatrice and Samuel A. Seaver Foundation.

## Author contributions

M.A., S.B., K.T. and A.R. designed the study. C.A. and M.A. undertook sample analysis. C.G. and M.A. conducted the statistical analysis. All authors contributed to acquisition of data, interpretation of results and writing of the manuscript.

## Additional information

**Competing interests:** The authors declare no competing financial interests.

**Publisher's note**: 

