## [Peer Review File · Nature Communications]

Reviewers' comments:

Reviewer #1 (Remarks to the Author):

This report summarizes the results from a sophisticated analysis of metals in deciduous teeth from twins with and without autism spectrum disorder (ASD). The authors draw conclusions on metals retention in the body based on a small number of subjects. The study is original, as the tooth analysis has not been used for such purpose before. However, the small number of subjects makes this case series less convincing. The approach is highly appropriate, although the authors overstate the importance of their study. The use of statistics seems appropriate, but it is not obvious how many cases were included in Figures 3 and 4. The discussion assumes that the exposure assessment and the timing of exposure are valid, and proper reservations are not considered. The conclusions likewise overstate what can be inferred from this study. The study would greatly improve if the subject numbers could be increased. That may be hard, but the current report refers to a case series only, and proper reservations are missing. My specific comments are as follows:

1. In line 71, the authors highlight the shortcoming of previous reports that represent "patient series rather than population-based samples". I think the present manuscript can best be characterized as a patient series due to the small number included from a large background population. However, the use of twins is a powerful advantage, which is then counterbalanced by the small numbers.
2. The laser ablation technique coupled to ICP is an interesting methodological development, but to say that it has been "rigorously validated" and that it "objectively reconstructs" past exposures is an exaggeration. Reference 12 does not provide justification for these statements. I would characterize the present study as explorative, given the fact that the exact timing of the metal incorporation in dental tissues is only partially known, and that incorporation in the teeth may not reflect the amount of the metals that passes through the blood-brain barrier.
3. The number of subjects included is not clear. Table 1 provides some numbers, e.g., 15 individuals belonging to ASD-discordant twin pairs with a total of 15 individuals. Results are then given for the ASD twin and the healthy co-twin, but without stating how they made up a total of 15 subjects - how many of each? Figure 2 shows data from three discordant twin pairs, but it is not clear how they were selected. The time scales are different, but it seems that exposures in twins from the same pair cover different time periods. Exposures among ASD-affected twins appear to be covered from about 100 days before birth to about 100 days after birth. Figures 3 and 4 rely on data from a total of six ASD twins, and exposures are modeled through to 30 weeks, i.e. about twice the duration covered in Figure 2. The Method section refers to 6 ASD twins and 12 healthy twin pairs (line 302), and 5 ASD concordant twins (line 308) (only one of each concordant twin pairs is shown in Figure 2).
4. The authors do not discuss whether the differences observed could be due to variations in dental tissue formation, but they refer to "discrete developmental periods" (line 196) without justification. Given that postnatal exposure for some subjects was covered for 200 days (concordant ASD twin at the bottom of Figure 2), while only 50 days for one healthy DZ twin, there appears to be sufficient difference between the assessments to require some reservations.
5. Studies of lead exposure have amply demonstrated that prenatal exposure is usually much lower than postnatal exposures, although the increased exposure may not happen until more than the 100-200 postnatal days covered by this study. Both members of the healthy MZ pair shown in Figure 2 had steep decreases in lead exposure right after birth, a decrease that is not in accordance with the half-life of lead in the body. A similar drop was seen in one more health twin, but not in the ASD patients. These drops in lead exposure, which I believe are implausible, are probably driving the significant difference shown in Figure 3.
6. Given that the focus is on exposures right after birth, it is notable that the report does not comment on breast-feeding (or bottle milk) as a source of metal exposures. Were any of the twin pairs discordant in regard to breast-feeding, e.g., due to hospitalization?
7. In the Methods section, the authors describe the teeth analyzed as "shed deciduous teeth".

Previous research has shown that lead retention in deciduous teeth varies by tooth type. Was this factor controlled, and could any differences related to tooth type be detected?

8. Manganese is an essential element, and its metabolism is affected by the binding to transferrin and for that reason probably also by iron status. The decreasing trends in dental manganese levels are interesting, but difficult to interpret, given the lack of knowledge of blood-manganese levels before and right after birth. Zinc shows much more stable levels. Given that differences were found also for other elements, I regard this an interesting pilot study that ought to be published, given that proper caveats can be added.

Reviewer #2 (Remarks to the Author):

A. Summary of the key results

This is an exploratory study of the possible association between levels of metal accumulation in deciduous teeth and autistic symptomatology which concludes that there may be an association.

B. Originality and interest:

The study features use of a twin cohort, prospective longitudinal follow-up, and a method in which metal concentration can be traced to developmental epochs prior to the usual onset of autistic symptoms, specifically the second and third trimester of pregnancy and the early postnatal years, all of which are potential advantages. Any confirmed association between metal exposure (or metabolism) and the development of autism would be of clinical interest, especially since exposures are preventable.

C. Data & methodology: validity of approach, quality of data, quality of presentation

The manuscript makes repeated reference to the sample being "population-based." This is very misleading. The sample is a small, highly-ascertained sub sample from a sub sample of two massive population-based studies. To illustrate but one monumental way in which it deviates from representativeness of the population, the proportion of MZ twins concordant for ASD in the reported sample is 50%, and the proportion of DZ twins concordant for ASD is 40%. This deviates radically from the authors' own population based data regarding MZ and DZ twin concordance. Thus, this sample might be more appropriately described as a convenience sample of a small number of families who donated baby teeth, some of which happen to be the teeth of twins.

The specific methods for metal ascertainment are not in my area of expertise and therefore I would defer to expert perspectives about their validity, but a fundamental question not explicitly addressed in the manuscript is the level of "independence" of serial measurements of metal accumulation over successive epochs. In other words, if latent class analysis or equivalent methods were applied to the ascertainment of metal accumulation over time in a given tooth sample, to what extent would a measurement at time x predict all other measurements (i.e. class assignment for a developmental metal accumulation trajectory)? Whether there exist contrasting developmental trajectories for individual subjects (in contrast to a scenario of high preservation of baseline individual differences over time-this is not specified by the authors) would seem to have significant implications for the manner in which correction for multiple testing should be applied (see below).

The method implemented for adjusting case-control differences for average control differences is skewed in favor of finding case-control differences since most of the control pairs were identical twins.

The manuscript does not report the method for establishing zygosity -if conducted by questionnaire rather than genotype, it could introduce another confound, since even a small number of misclassifications (which are common) on the basis of zygosity questionnaires could affect the interpretation of the data.

D. Appropriate use of statistics and treatment of uncertainties

I will leave it to a statistical reviewer to specify just how these results should be most appropriately chance corrected (the manuscript describes Bonferonni correction but not exactly how this was done or for what number of tests the results were corrected). It is difficult to conceive of any result that would survive the number of corrections necessary for this particular design at a sample size as low as 6: ten metals, numerous developmental intervals, three different approaches to the data, and the possibility / likelihood that the successive measurements are not independent of one another.

E. Conclusions: robustness, validity, reliability

Given the substantial ascertainment issues, the very small sample size, the number of metals and developmental epochs tested, and the unanswered question about whether a single point in time specifies an infant trajectory for metal accumulation for any given metal, a study involving 23 twin pairs seems seriously underpowered to test the many developmental hypotheses in play. Given the nature of the case-control selection of the sample, even the categorical and quantitative analyses cannot be construed as independent of one another since any case-control difference that arises by chance would be likely to translate to a quantitative bivariate correlation. Defensible assertion of a significant association between metal accumulation and ASD would require a substantially larger sample, and would optimally differentiate environmental versus genetic mediation of the relationship between metal levels and ASD symptoms.

F. Suggested improvements: experiments, data for possible revision

A focus of discordant MZ twins would be particularly interesting and informative, however only 3 such pairs are available in this data set.

G. References: appropriate credit to previous work?

References appear appropriate

H. Clarity and context: lucidity of abstract/summary, appropriateness of abstract, introduction and conclusions

Repeated assertion of the sample being population-based should be revised.

Reviewer #3 (Remarks to the Author):

Review for Nature communications: Foetal and Postnatal Metal Dysregulation in Autism

This paper uses data acquired from tooth matrix in a small sample of twins ascertained through The Roots of Autism and ADHD Twin Study in Sweden to test for differences in the presence of various trace metals at different time points in early development. The authors highlight that, while autism is a strongly genetically influenced disorder, current evidence nevertheless points to a role for environmental factors and gene-environmental interaction. The authors assess a total of 10 trace elements but focus particularly on lead, manganese and zinc, for which they argue there are particular reasons to implicate as possible etiological factors in developmental disorders. The findings are based

on a very small sample but are nevertheless suggestive of complex differential patterns of exposure to these three trace elements during the early developmental period. The present findings are intriguing and will be worthy of follow up.

I have no expertise in the measurement of trace metals and cannot comment on the appropriateness of the methods applied in this paper. The authors point to the benefits of using tooth matrix in being able to get a longitudinal picture of trace metal exposure. Assuming their assertions are correct, this should prove a powerful technique for retrospectively examining variation during the early developmental period. Another strength of the present paper is the use of twin data. The authors have already published a descriptive paper on the present cohort (RATSS) and the diagnostic measures used to classify individuals as having an autism spectrum disorder (ASD) are gold standard. However, looking at both table 1 and supplementary figure 1, there are several slightly troubling features of the sample. The first is the high IQ in those with ASD. Most current studies are reporting a mean IQ in the low to mid 70's while here there appears to be no difference in IQ from the general population. Furthermore, unless there has been a large secular change in the proportion of MZ and DZ twins, I would have expected the population rates to be approximately 1/3 MZ and 2/3 DZ, the latter being equally divided between opposite- and same-sex pairs. This small sample appears to have an overrepresentation of MZ twins and I was uncertain whether opposite-sex pairs had been included. This is important because the authors assert the strength of the present study is its derivation from the general population. Therefore we need to know whether there are any ascertainment biases. The authors could address this by augmenting the information in supplementary figure 1. For example in the top line in the table, it would be helpful to know the numbers coming into RATSS from the different ascertainment strategies (I note that the use of advertisements, the second box, often skews ascertainment to higher rates of MZ twins and concordant pairs). The important element here is really to understand how representative the sample is.

A second element that would be helpful is to test whether the unaffected twins in discordant pairs are similar in their metal profiles to the concordant unaffected twins. This raises the questions of requiring more information on what is meant by 'discordance'. There is much information from autism twin studies previously indicating that discordance for meeting the diagnostic threshold of ASD may be relatively high, e.g. 30%, but many of the co-twins not meeting diagnostic threshold showed some degree of impairment. Examining the data in table 1 suggests this is not the case in RATSS, as the unaffected co-twins show ADOS and SRS scores similar to or even slightly lower than the concordant unaffected twins. I presume they were selected on this basis and it would be helpful if the authors clarified this in the methods. Given this, the patterns of trace metal exposure in the discordant co-twins can help to shed light on the causal nature of any risk, i.e. to differentiate between the role of environmental exposure and gene-environment interplay in relation to exposure where, in the first model one would expect the discordant co-twins to be similar to the concordant unaffected twins while in the later model they could be intermediate between their affected co-twin and the unaffected twin pairs. I recognize the numbers are small but this would nevertheless be of interest.

I found figure 2 somewhat confusing. In panel B I was expecting to see two series of graphs for MZ and DZ discordant pairs and could not understand the two sets of panels for MZ discordant pairs. Similarly, in panel C, I was expecting to see separate graphs for MZ concordant and DZ pairs concordant for autism while there are only MZ concordant pairs.

Figure 4 looks at profiles for lead, manganese and zinc according to SRS and ADOS severity across the entire sample of unaffected and affected twins. However, I suspect that the two distributions of scores are not continuous and that there is a lack of intermediate scores on both measures because of the nature of the sample selection. If this is correct I think the statistical approach taken is probably not valid.

Finally, the authors suggest or state at several places in the manuscript that the findings may indicate a causal mechanism because the timing of the abnormalities they observe is prior to the onset of autism symptoms. This is a weak argument as we know there are a number of Mendelian disorders, e.g. Huntington disease, muscular dystrophy, caused by a single genetic mutation in which the phenotypic manifestations take many years to be observed.

In summary, this is an intriguing paper using novel methodology to suggest a possible etiological role for abnormalities in trace metals in autism. The manuscript has a number of strengths, but also limitations and I have focussed on the later in this review. Nevertheless, I think it is of sufficient importance to be given serious consideration.

Reviewer #4 (Remarks to the Author):

I was asked to review this paper due to my expertise in laser ablation inductively coupled mass spectrometry (LA-ICPMS). As such I will be limiting my comments to that aspect of the paper and to the best of my knowledge and ability will try to address the impact of the LA-ICP-MS on the conclusions. The biggest issue I have is the almost complete lack of experimental detail. It consists of a very cursory description of the equipment used with no information at all about laser parameters such as spot size, fluence, repetition rate or any similar information about the ICP-MS. Two references are given to earlier work but I only had access to one, Austin C, et al. Barium distributions in teeth reveal early-life dietary transitions in primates. *Nature* 498, 216-219 (2013). This reference contained slightly more detail about the laser, but it was a completely different manufacturer and type, New Wave, Nd:Yag at 213 nm vs ESI, ArF at 193 nm (the wavelength was not given, it is my own knowledge that filled that in. A different type of ICP-MS was used as well. As a result this reference gives me little to go on. There was yet another reference given here, but in a dental journal that I do not have access to.

In spite of this lack of detail, the data shown in figure 2 do seem to support the stated result of observed differences in the temporal accumulation of Pb, Mn and Zn between discordant pairs of twins and the relative similarities among concordant pairs. I would recommend this paper for publication with the addition of more detail on the LA-ICP-MS method as described above.

We are grateful to the reviewers for the very thoughtful critique of our manuscript and are pleased to say that we have addressed all the concerns raised. We would like to highlight two major improvements we have made – an increase in sample size of 50% and improved statistical analysis (detailed below). Our results remain qualitatively robust confirming our initial findings (zinc and manganese are lower in ASD cases and lead is higher).

- 1. Sample Size:** During the past several months since initial submission of our manuscript we have devoted substantial resources and full-time personnel to increase the number of samples collected in our primary RATSS population in Sweden. We are pleased to say that we have increased sample size by 50% (from $n=51$ to 76). We were able to do this by contacting every participant within the tooth shedding ages. Our sample now represents 26.5% of the total RATSS population, 50% of the RATSS study participants who are within the ages of tooth shedding, and in general RATSS has a substantial subsample (~11.3%) of all discordant twins in Sweden in this age range (Lichtenstein et al., 2010). We do not anticipate any more participants shedding teeth to be available in the near future.

Furthermore, we undertook targeted convenience recruitment at an autism clinical center in New York, USA. We were able to recruit one ASD discordant twin pair. Data from this pair supports our primary findings and is included in Supplemental Material.

- 2. Statistical Analysis:** We have refined our primary statistical approach (distributed lag models (DLMs)) to include *two levels* of stringent corrections. This conveys to the reader our results at multiple levels – when correcting for within twin correlations and when correcting for multiple comparisons. Our results remain statistically significant even after we have undertaken these strict corrections. We choose to be conservative when presenting our results and avoid false positives, since these findings have important implications for the better understanding of ASD, in our opinion.

Point-by-point responses to reviewer comments are given below. Please note line numbers noted are for the unmarked copy of the manuscript (where all Track Changes have been removed).

Reviewer #1 (Remarks to the Author):

This report summarizes the results from a sophisticated analysis of metals in deciduous teeth from twins with and without autism spectrum disorder (ASD). The authors draw conclusions on metals retention in the body based on a small number of subjects. The study is original, as the tooth analysis has not been used for such purpose before. However, the small number of subjects makes this case series less convincing. The approach is highly appropriate, although the authors overstate the importance of their study. The use of statistics seems appropriate, but it is not obvious how many cases were included in Figures 3 and 4. The discussion assumes that the exposure assessment and the timing of exposure are valid, and proper reservations are not considered. The conclusions likewise overstate what can be inferred from this study. The study would greatly improve if the subject numbers could be increased. That may be hard, but the current report refers to a case series only, and proper reservations are missing.

Authors' response: We thank Reviewer 1 for the in-depth review and very helpful suggestions.

- We have added numbers of cases and controls to each figure as suggested. In our original submission, we had shown the case and control numbers along with zygosity in Supplementary Figure 1, but have now brought that figure to the main manuscript (now Figure 1) to clarify our sample and study design.
- In keeping with the reviewer's suggestion, we have tempered our conclusions and highlighted the limitations of our study, including limited sample size. Additional details on this point are below.
- As highlighted above, during the past several months since initial submission of our manuscript we have devoted substantial resources and full-time personnel to increase the number of samples collected in our primary RATSS population in Sweden. We are pleased to say that we have increased sample size by 50% (from $n=51$ to 76). We were able to do this by contacting every participant within the tooth shedding ages. Our sample now represents 26.5% of the total RATSS population, 50% of the RATSS study participants who are within the ages of tooth shedding, and in general RATSS has a substantial subsample (~11.3%) of all discordant twins in Sweden in this age range (Lichtenstein et al.,

2010). We do not anticipate any more participants shedding teeth to be available in the near future.

My specific comments are as follows:

1. In line 71, the authors highlight the shortcoming of previous reports that represent "patient series rather than population-based samples". I think the present manuscript can best be characterized as a patient series due to the small number included from a large background population. However, the use of twins is a powerful advantage, which is then counterbalanced by the small numbers.

Authors' response: In agreement with the suggestion of the reviewers, we have removed all mention of our study being "population-based". We agree that the overall small sample size is a limitation of our study and it is a non-random sample of the base population, but we would like to differentiate our study from others that had used a convenience sample (i.e. recruiting from a clinic or hospital). While our sample is small it includes 26.5% of the total population-based RATSS study in Sweden. Importantly, if we consider children of tooth shedding age, our sample is 50% of the total RATSS population.

To clarify this distinction between our work and other studies using clinic-based samples, we have changed our text from the original statement. Please see line 83 where we state *"We enrolled 26.5% of the base study population of the Roots of Autism and ADHD Twin Study in Sweden (RATSS) through a non-random selection process (Fig. 1). RATSS applies deep phenotyping on a significant subset (currently 11.3%) of all ASD discordant twins in Sweden in the specified age range.¹³ Importantly, since the RATSS study only recruits participants older than 8 years, and children only shed teeth until the age of 12 years, the study we report here represents 50% of the RATSS base population who are of tooth shedding age."*

Lines 260: We have stated that our sample is "small" and "non-random" which is a limitation of our study.

2. The laser ablation technique coupled to ICP is an interesting methodological development, but to say that it has been "rigorously validated" and that it "objectively reconstructs" past exposures is an exaggeration. Reference 12 does not provide justification for these statements. I would characterize the present study as explorative, given the fact that the exact timing of the metal incorporation in dental tissues is only partially known, and that incorporation in the teeth may not reflect the amount of the metals that passes through the blood-brain barrier.

Authors' response: We agree with the reviewer that concentrations of lead in teeth are not a direct measure of the kinetics of lead across the blood brain barrier. However, we would like to reassure the reviewer that we have indeed undertaken validation studies comparing the tooth biomarkers with blood assays and also environmental data in humans and animals. We have provided references to the 5 papers that we have published on validation studies and added a paragraph describing the validation approach. In addition to our epidemiologic studies, we undertook a rat study (Arora et al. 2015) where we compared tooth levels of lead with concentrations in brain, blood kidney, liver and bone. Tooth lead levels showed strong correlations with brain levels ($r = 0.91$).

Please see Line 325 where we state that, *"We have undertaken detailed validation of these tooth-matrix assays against other metal biomarkers and measures in environmental samples. For example, we validated prenatal lead levels in teeth with maternal blood lead measured during the second and third trimesters of pregnancy, and birth and childhood measures of tooth lead with lead concentrations in umbilical cord blood and serial childhood blood measures, respectively [Arora et al. 2014]. We have similarly validated metal levels in teeth against environmental samples (house dust concentrations and distance to exposure source, for example) [Arora et al. 2012, Gunier et al. 2013], and also undertaken detailed animal studies with controlled exposures (Arora et al. 2015; Austin et al. 2013)."*

3. The number of subjects included is not clear. Table 1 provides some numbers, e.g., 15 individuals belonging to ASD-discordant twin pairs with a total of 15 individuals. Results are then given for the ASD twin and the healthy co-twin, but without stating how they made up a total of 15 subjects - how many of each? Figure 2 shows data from three discordant twin pairs, but it is not clear how they were selected. The time scales are different, but it seems that exposures in twins from the same pair cover different time periods. Exposures among ASD-affected twins appear to be covered from about 100 days before birth to about 100 days after birth. Figures 3 and 4 rely on data from a total of six ASD twins, and exposures are modeled through to 30 weeks, i.e. about twice the duration covered in Figure 2. The Method section refers to 6 ASD twins and 12

healthy twin pairs (line 302), and 5 ASD concordant twins (line 308) (only one of each concordant twin pairs is shown in Figure 2).

Authors' response: We thank the reviewer for this detailed feedback. We have made edits to clarify the sample sizes in each analysis.

- Number of subjects: We have made a new figure (now Figure 1) to show the number of subjects, zygosity and ASD status in the whole study.
- The main purpose of Figure 2 was to provide the reader with examples of the trends in metal distribution at an individual level. For clarity and brevity, we showed examples using data from 2 x MZ discordant, 1 x DZ discordant and 1 x MZ non-ASD/control. If we attempted to show all the participants, the figure becomes extremely crowded. In Figures 3 and 4, we have results for the whole study population. If the editor and reviewer wish, we can provide individual level graphs for all participants with the clarification that this would be a very large figure and substantially increase the length of the manuscript or online supplement. We do provide individual level data on all participants in Supplementary Figure 3.
- Time scales are different because different teeth develop over different time periods (summarized in table below), but all have a prenatal and postnatal component. In Figure 2 we had truncated some of the values to focus on the developmental times where (i) we had data from both siblings in a twin pair, and (ii) there was a divergence in metal concentrations between the siblings. For our statistical models, in Figure 3 and 4, we use the full data.
- We have added case and control numbers to each of the Figures in the Results section.

4. The authors do not discuss whether the differences observed could be due to variations in dental tissue formation, but they refer to "discrete developmental periods" (line 196) without justification. Given that postnatal exposure for some subjects was covered for 200 days (concordant ASD twin at the bottom of Figure 2), while only 50 days for one healthy DZ twin, there appears to be sufficient difference between the assessments to require some reservations.

Authors' response: The reviewer has raised an important point about the impact of the biology of teeth on our measures of metals. We apologize for not clarifying this important issue previously. We have undertaken several studies (in animals and humans) to address this. The key studies are listed under response to Point 2 above, but the main aspects are:

- Different teeth types (incisors, canines and molars) develop at different times, but our methodology of mapping the temporal lines (growth rings) takes that into account. We rely on the neonatal/birth line in each tooth as a reference point. In this manner, we are able to identify different prenatal and postnatal time points in any tooth type.
- In the table below we have shown developmental timing for primary mineralization, which allows us to reconstruct the early life metal exposure. When siblings in a twin pair provide us with different types of teeth we are able to compare the common time periods between the two tooth types. For example, if one twin provides a canine and the other provides a first molar, we are able to investigate difference in metal uptake between approx. day 120 before birth and day 300 after birth.
- If any tooth has a crack or any other damage, we do not analyze that part which is why sometimes the time periods covered are smaller than those shown in the table below.
- These methods and the metal concentrations obtained by this assay have been validated in human and animal studies where tooth measures were compared to blood and bone concentrations and also environmental measures (levels in house dust, diet, and other sources of exposure). Please see Point 2 above for additional detail.

5. Studies of lead exposure have amply demonstrated that prenatal exposure is usually much lower than postnatal exposures, although the increased exposure may not happen until more than the 100-200 postnatal days covered by this study. Both members of the healthy MZ pair shown in Figure 2 had steep decreases in lead exposure right after birth, a decrease that is not in accordance with the half-life of lead in the body. A similar drop was seen in one more health twin, but not in the ASD patients. These drops in lead exposure, which I believe are implausible, are probably driving the significant difference shown in Figure 3.

Authors' response: We thank the reviewer for making these two very important points.

1. There is mixed data on prenatal vs postnatal lead exposures. In some populations, prenatal exposures are lower, as the reviewer correctly points out, but the converse has also been shown. As an example, below we show data from a study where umbilical cord blood lead (which represents fetal circulating lead) is higher than infant blood lead measured at 1 month post-birth.

Source: Table 1 of Environ Health Perspect. 2004 Oct; 112(14): 1381–1385.

Biomarker of lead exposure	No.	Mean ± SD	Minimum	Maximum
At delivery				
Maternal blood lead (µg/dL)	251	8.7 ± 4.2	2.1	23.7
Umbilical cord lead (µg/dL)	222	6.7 ± 3.6	1.2	26.3
At 1 month postpartum				
Breast milk lead (µg/L)	255	1.5 ± 1.2	0.3	8.0
Maternal blood lead (µg/dL)	255	9.4 ± 4.5	1.8	29.9
Maternal patella lead (µg/g)	246	15.3 ± 15.0	< 1	67.2
Maternal tibia lead (µg/g) ^a	250	10.0 ± 10.4	< 1	76.6
Infant blood lead (µg/dL)	255	5.5 ± 3.0	1	23.1

2. As the reviewer mentions, the rapid decline of lead in dentine post-birth is indeed striking but we would like to clarify that the kinetics of lead in a newborn are very different to that of an adult. Given the rapid growth of organs, including the skeletal system that accumulates much of the body's lead, the higher clearance of lead from infant blood has been proposed before. The table below summarizes other studies that have shown the half-life of infant lead is shorter than what is seen in adults. The overarching issue here is that the kinetics of lead in human blood are mostly based on studies of adults and not newborns. While it has been hypothesized that infant blood will clear lead faster than adults (and our data is in agreement with this), exact kinetics are still poorly understood. However, we agree with the reviewer that this is an issue that must be highlighted as a limitation of our study and we have thus included a statement to clarify this point.

We have added the following statement to line 273: "Our study also highlights a need to study the kinetics of metal mixtures during foetal and early postnatal development as we noted substantial shifts in metal levels between the pre-and postnatal periods. Most studies on the clearance and half-life of metals in humans are undertaken on adults and not newborns."

Study Group and Exposure	Half-Life, days	Comments (References)
Infants, middle class; ambient exposure	-	Blood lead very unstable for first 20 mo Rabinowitz et al., 1984)
Infants, middle class; ambient low exposure	5.6	Re-analysis of Ziegler et al. et al. (1978) data; mean-time 8 days (half-life, 5.6 days) (Duggan, 1983)
Infants, low socioeconomic status; heavy ambient exposure	ca. 300	Reflects high body burden plus in utero uptake in urban setting (Succop et al., 1987)
Low-socioeconomic-status children of battery workers; secondary exposure	-	Rank order of group preserved over 5 yr; $r = 0.74$ (Schroeder et al., 1985)
General U.S. child population; varied exposure (children older than 2 yrs)	-	Regression analyses of NHANES blood lead data showed 30-day (best-fit) lag with lead source (Schwartz et al., 1985; Annest and Mahaffey, 1984)
School-age English children; low exposure	-	Two blood lead sets, 20 mo apart; rank order preserved (Landsdown et al., 1986)
U.S. children, 4–12 yrs old; increased ambient exposure	-	Rank order of serial blood lead measures generally preserved (David et al., 1982)

Source: Table 4-3. *Studies of Kinetic Behavior of Lead in Blood of Children. Measuring Lead Exposure in Infants, Children and Other Sensitive Populations. National Research Council (US). Committee on Measuring Lead in Critical Populations. Washington (DC): National Academies Press (US); 1993.*

6. Given that the focus is on exposures right after birth, it is notable that the report does not comment on breast-feeding (or bottle milk) as a source of metal exposures. Were any of the twin pairs discordant in regard to breast-feeding, e.g., due to hospitalization?

Authors' response: The tooth-matrix biomarkers capture internal uptake of metals from all sources including breastfeeding, so differences in metal exposure from breastfeeding would be reflected in our measures. However, the reviewer is correct that we are not identifying specific sources of exposure, which we have now noted as a limitation of our study.

Furthermore, to address the reviewer's concerns, we did contact our participants to collect data on breastfeeding and no discordancy was apparent in the responders between twins.

On line 271 we have added the statement: *"Our tooth-biomarkers measure uptake of metals from all sources but do not distinguish specific sources such as breastfeeding."*

7. In the Methods section, the authors describe the teeth analyzed as "shed deciduous teeth". Previous research has shown that lead retention in deciduous teeth varies by tooth type. Was this factor controlled, and could any differences related to tooth type be detected?

Authors' response: In older studies of the late 1970s and 1980s, whole teeth were pulverized for metal analysis and for those older analytical methods tooth type was relevant because of the differences in tooth anatomy. Our method measures metals at a microspatial resolution and is not affected by tooth type because we sample the growth rings in the same location relative to the birth line in every tooth. Nonetheless, in the revised manuscript we also adjust for tooth type. Tooth type was not a statistically significant covariate and had no effect on the results.

8. Manganese is an essential element, and its metabolism is affected by the binding to transferrin and for that reason probably also by iron status. The decreasing trends in dental manganese levels are interesting, but difficult to interpret, given the lack of knowledge of blood-manganese levels before and right after birth. Zinc shows much more stable levels. Given that differences were found also for other elements, I regard this an interesting pilot study that ought to be published, given that proper caveats can be added.

Authors' response: We thank the reviewer for recognizing this work as worthy of publication. In keeping with their suggestion, we have added several statements to the limitations of our study, and also expanded our sample size in the Swedish RATSS study. Furthermore, we sourced a discordant twin pair from another study in the US to see if we found similar trends as in our main study. While this one additional pair does not provide a full validation and some limitations of our study cannot be completely eliminated, we hope that with these efforts, we are providing the readers with a balanced view of our findings.

Reviewer #2 (Remarks to the Author):

A. Summary of the key results

This is an exploratory study of the possible association between levels of metal accumulation in deciduous teeth and autistic symptomatology which concludes that there may be an association.

B. Originality and interest:

The study features use of a twin cohort, prospective longitudinal follow-up, and a method in which metal concentration can be traced to developmental epochs prior to the usual onset of autistic symptoms, specifically the second and third trimester of pregnancy and the early postnatal years, all of which are potential advantages. Any confirmed association between metal exposure (or metabolism) and the development of autism would be of clinical interest, especially since exposures are preventable.

Authors' response: We thank the reviewer for their encouraging appraisal and for the valuable feedback, which has helped us improve our work. We agree that this work will set the scene for with future studies that will lead to clinically relevant outcomes.

C. Data & methodology: validity of approach, quality of data, quality of presentation

The manuscript makes repeated reference to the sample being "population-based." This is very misleading. The sample is a small, highly-ascertained sub sample from a sub sample of two massive population-based studies. To illustrate but one monumental way in which it deviates from representativeness of the population, the proportion of MZ twins concordant for ASD in the reported sample is 50%, and the proportion of DZ twins concordant for ASD is 40%. This deviates radically from the authors' own population based data regarding MZ and DZ twin concordance. Thus, this sample might be more appropriately described as a convenience sample of a small number of families who donated baby teeth, some of which happen to be the teeth of twins.

Authors' response: Both Reviewer 1 and 2 have raised this concern. We agree that the small sample size is a limitation of our study, and the children who donated teeth are not necessarily representative of the whole Swedish twin population. We have removed all mention of our study being "population-based". However, we would like to differentiate our study from others that had used a convenience recruitment strategy (such as recruiting from a clinic or hospital). While our sample is small, it is approximately 26.5% of the total population-based RATSS study in Sweden. If we consider children within the age of tooth shedding (6 to 12 years), our study sample is 50% of the total RATSS population, as well as about 11.3% of the total population of discordant twins in the age range (compare to Lichtenstein et al., 2010).

To clarify this distinction between our work and other studies, we have changed our text from the original statement Please see line 82 where we state *"We enrolled 26.5% of the base study population of the Roots of Autism and ADHD Twin Study in Sweden (RATSS) through a non-random selection process (Fig. 1). RATSS applies deep phenotyping on a significant subset (currently 11.3%) of all ASD discordant twins in Sweden in the specified age range.¹³ Importantly, since the RATSS study only recruits participants older than 8 years, and children only shed teeth until the age of 12 years, the study we report here represents 50% of the RATSS base population who are of tooth shedding age."*

The specific methods for metal ascertainment are not in my area of expertise and therefore I would defer to expert perspectives about their validity, but a fundamental question not explicitly addressed in the manuscript is the level of "independence" of serial measurements of metal accumulation over successive epochs. In other words, if latent class analysis or equivalent methods were applied to the ascertainment of metal accumulation over time in a given tooth sample, to what extent would a measurement at time x predict all other measurements (i.e. class assignment for a developmental metal accumulation trajectory)? Whether there exist contrasting developmental trajectories for individual subjects (in contrast to a scenario of high preservation of baseline individual differences over time-this is not specified by the authors) would seem to have significant implications for the manner in which correction for multiple testing should be applied (see below).

Authors' response: The reviewer has raised a very important point. We address this under point D below and

also detail it in the Supplemental Materials. Overall, we have given much attention to these issues in our statistical analysis and also conveyed the results to the reader at three levels so that they may easily appreciate how exposure-disease associations are affected by each statistical procedure (see Figure 4-6 in main manuscript).

The method implemented for adjusting case-control differences for average control differences is skewed in favor of finding case-control differences since most of the control pairs were identical twins.

Authors' response: We agree with the reviewer that such an imbalance may have affected our initial results. However, as we increased our sample size from 51 to 76, this imbalance has now diminished. We have more DZ controls pairs than identical twins. Our results still hold indicating that the zygosity of the control twins was not introducing a major bias in our study. Nonetheless, we include zygosity as a covariate in our analysis to adjust for its effects.

The manuscript does not report the method for establishing zygosity-if conducted by questionnaire rather than genotype, it could introduce another confound, since even a small number of misclassifications (which are common) on the basis of zygosity questionnaires could affect the interpretation of the data.

Authors' response: the zygosity is based on DNA analysis. This information is included the Supplemental Material where we state that:

“Zygosity was determined by genotyping of saliva or whole-blood derived DNA using standard methods. The genotyping was done using Infinium Human-CoreExome chip (Illumina Inc. USA). The estimating identity by descent was analyzed using the PLINK software (v1.07)⁵ after quality control and removal of SNPs with minor allele frequency less than 0.05 within the samples. All pairs of DNA samples showing $\hat{\pi} \geq 0.99$ were considered as monozygotic pairs. For few pairs, a short tandem repeat kit (Promega Powerplex 21) was used to determine the zygosity”.

D. Appropriate use of statistics and treatment of uncertainties

I will leave it to a statistical reviewer to specify just how these results should be most appropriately chance corrected (the manuscript describes Bonferroni correction but not exactly how this was done or for what number of tests the results were corrected). It is difficult to conceive of any result that would survive the number of corrections necessary for this particular design at a sample size as low as 6: ten metals, numerous developmental intervals, three different approaches to the data, and the possibility / likelihood that the successive measurements are not independent of one another.

Authors' response: Distributed lag models (DLMs) treat the exposure at all time points as a set, taking into account the inter-correlatedness of serial exposure measures. This is the primary reason we have used this statistical approach. DLMs are markedly different to, for example, having 40 regression models at 40 different time points, which would create issues of multiple comparisons and false assumptions of independence between successive measurements.

Although most studies using DLMs do not apply any corrections other than 95% confidence intervals, we prefer to take a conservative approach and apply two additional levels of corrections. We have also presented our graphs in a manner that the reader can readily see the effects of each correction.

First we construct 95% confidence bands at each time point to account for the variability in our model (gray bands in Figures 4-6). We then construct wider bands that take into account intra-pair correlations (blue bands in Figures 4 – 6). Finally, we further penalize our results by applying a Holm-Bonferroni correction for a comparison every 5th week (11 comparisons).

We have detailed each step of this statistical analysis in the Methods section and Supplementary Materials.

We would like to stress that this is the most stringent correction (i.e. the most conservative application) of DLMs that we know of. Most other studies only report 95% confidence bands without any additional penalization. For examples, see PMID: 26176842, and PMID: 27261529.

E. Conclusions: robustness, validity, reliability

Given the substantial ascertainment issues, the very small sample size, the number of metals and developmental epochs tested, and the unanswered question about whether a single point in time specifies an infant trajectory for metal accumulation for any given metal, a study involving 23 twin pairs seems seriously underpowered to test the many developmental hypotheses in play. Given the nature of the case-control selection of the sample, even the categorical and quantitative analyses cannot be construed as independent of one another since any case-control difference that arises by chance would be likely to translate to a quantitative bivariate correlation. Defensible assertion of a significant association between metal accumulation and ASD would require a substantially larger sample, and would optimally differentiate environmental versus genetic mediation of the relationship between metal levels and ASD symptoms.

Authors' response: We thank the reviewer for highlighting these limitations. We have taken their comments to heart and invested considerable resources to contact every participant in RATSS cohort. Since the original submission, we have increased our sample size by almost 50% (N=51 to 76) and also recruited one additional discordant pair from an autism clinic (analyzed separately to the main study). We would like to clarify that recruiting ASD discordant twin pairs is very difficult and, even though our study is small, in our opinion it provides useful information. We recognize that larger sample sizes are required to differentiate genetic and non-genetic contributions in understanding the relation between metals and ASD and have added this point to our Discussion (please see Lines 265-267).

F. Suggested improvements: experiments, data for possible revision

A focus of discordant MZ twins would be particularly interesting and informative, however only 3 such pairs are available in this data set.

G. References: appropriate credit to previous work?
References appear appropriate

H. Clarity and context: lucidity of abstract/summary, appropriateness of abstract, introduction and conclusions

Repeated assertion of the sample being population-based should be revised.

Authors' response to F, G and H: We have removed the mention of population-based sample and noted that it is a non-random sample of participants who could donate teeth.

We agree that comparison of MZ discordant twins would be interesting but these are extremely rare and having 3 such pairs is not enough to undertake any advanced statistical analysis. We have noted in the Discussion that the small sample is a limitation of our study.

Reviewer #3 (Remarks to the Author):

Review for Nature communications: Foetal and Postnatal Metal Dysregulation in Autism

This paper uses data acquired from tooth matrix in a small sample of twins ascertained through The Roots of Autism and ADHD Twin Study in Sweden to test for differences in the presence of various trace metals at different time points in early development. The authors highlight that, while autism is a strongly genetically influenced disorder, current evidence nevertheless points to a role for environmental factors and gene-environmental interaction. The authors assess a total of 10 trace elements but focus particularly on lead, manganese and zinc, for which they argue there are particular reasons to implicate as possible etiological factors in developmental disorders. The findings are based on a very small sample but are nevertheless suggestive of complex differential patterns of exposure to these three trace elements during the early developmental period. The present findings are intriguing and will be worthy of follow up.

I have no expertise in the measurement of trace metals and cannot comment on the appropriateness of the methods applied in this paper. The authors point to the benefits of using tooth matrix in being able to get a longitudinal picture of trace metal exposure. Assuming their assertions are correct, this should prove a

powerful technique for retrospectively examining variation during the early developmental period. Another strength of the present paper is the use of twin data. The authors have already published a descriptive paper on the present cohort (RATSS) and the diagnostic measures used to classify individuals as having an autism spectrum disorder (ASD) are gold standard. However, looking at both table 1 and supplementary figure 1, there are several slightly troubling features of the sample. The first is the high IQ in those with ASD. Most current studies are reporting a mean IQ in the low to mid 70's while here there appears to be no difference in IQ from the general population. Furthermore, unless there has been a large secular change in the proportion of MZ and DZ twins, I would have expected the population rates to be approximately 1/3 MZ and 2/3 DZ, the latter being equally divided between opposite- and same-sex pairs. This small sample appears to have an overrepresentation of MZ twins and I was uncertain whether opposite-sex pairs had been included. This is important because the authors assert the strength of the present study is its derivation from the general population. Therefore we need to know whether there are any ascertainment biases. The authors could address this by augmenting the information in supplementary figure 1. For example in the top line in the table, it would be helpful to know the numbers coming into RATSS from the different ascertainment strategies (I note that the use of advertisements, the second box, often skews ascertainment to higher rates of MZ twins and concordant pairs). The important element here is really to understand how representative the sample is.

Authors' response: We thank the reviewer for raising this important point. We are showing the average IQ for ASD discordant, ASD concordant and non-ASD twin pairs in Table 1. As the reviewer points out there is no substantial difference in the average IQ within the groups. However within our ASD cases, 26.3% have IQ lower than 70 (6 out of 19 individuals), which is similar to the reported percentage of ASD cases with intellectual disability in the Swedish population (25.6%, Idring et al. 2015, J Autism Dev Disord; 45:1766–1773). Additionally, large ASD cohorts both from US and Canada have shown that the average IQ of the affected children is within normal range, similar to our study (Tammimies et al. 2015; JAMA; 314(9):895-903 and Iossifov et al. 2014, Nature Nov 13;515(7526):216-21).

A second element that would be helpful is to test whether the unaffected twins in discordant pairs are similar in their metal profiles to the concordant unaffected twins. This raises the questions of requiring more information on what is meant by 'discordance'. There is much information from autism twin studies previously indicating that discordance for meeting the diagnostic threshold of ASD may be relatively high, e.g. 30%, but many of the co-twins not meeting diagnostic threshold showed some degree of impairment. Examining the data in table 1 suggests this is not the case in RATSS, as the unaffected co-twins show ADOS and SRS scores similar to or even slightly lower than the concordant unaffected twins. I presume they were selected on this basis and it would be helpful if the authors clarified this in the methods. Given this, the patterns of trace metal exposure in the discordant co twins can help to shed light on the causal nature of any risk, i.e. to differentiate between the role of environmental exposure and gene -environment interplay in relation to exposure where, in the first model one would expect the discordant co-twins to be similar to the concordant unaffected twins while in the later model they could be intermediate between their affected co-twin and the unaffected twin pairs. I recognize the numbers are small but this would nevertheless be of interest.

Authors' response: We completely agree with this reasoning and the importance of the twin sample composition. The recommended information has been added to the revision and the point about representativeness and MZ/DZ ration has been clarified. We included the percentage of twins recruited from different sources in the new Figure 1. We also pointed out in the text that although our recruitment is population-based, the twin sample included in this study is non-random as we can only include twin pairs able to provide deciduous teeth and are willing to donate them for the research study. We include opposite-sex DZ pairs. Two opposite-sex pairs are included (clarified in page 5, line 104). Although, twins were also recruited via advertisement, we know that >80% of all of our twins included are from twin registries in Sweden including the population-based CATSS cohort, and represents about 11.3% of all ASD discordant pairs in the age range in Sweden (compare Lichtenstein et al., 2010).

I found figure 2 somewhat confusing. In panel B I was expecting to see two series of graphs for MZ and DZ discordant pairs and could not understand the two sets of panels for MZ discordant pairs. Similarly, in panel C, I was expecting to see separate graphs for MZ concordant and DZ pairs concordant for autism while there are only MZ concordant pairs.

Authors' response: The purpose of Figure 2 (now Figure 3) was to show: (a) typical patterns in the distribution of metals across developmental times, and (b) how cases and controls diverged. We showed two MZ discordant pairs because that is the most important component of our study. We attempted to make a figure as per the reviewer's suggestion but unfortunately, such a figure would be very long extending across several pages. We do provide individual level data on all participants in Supplementary Figure 3.

Figure 4 looks at profiles for lead, manganese and zinc according to SRS and ADOS severity across the entire sample of unaffected and affected twins. However, I suspect that the two distributions of scores are not continuous and that there is a lack of intermediate scores on both measures because of the nature of the sample selection. If this is correct I think the statistical approach taken is probably not valid.

Authors' response: We would like to reassure the reviewer that the ADOS-2 and SRS-2 scores are distributed across the entire range. As expected there are more in the lower range of ADOS-2 as we have a substantial number of non-ASD controls. Below we have plotted the frequency of ADOS and SRS scores across our study sample.

Finally, the authors suggest or state at several places in the manuscript that the findings may indicate a causal mechanism because the timing of the abnormalities they observe is prior to the onset of autism symptoms. This is a weak argument as we know there are a number of Mendelian disorders, e.g. Huntington disease, muscular dystrophy, caused by a single genetic mutation in which the phenotypic manifestations take many years to be observed.

Authors' response: In keeping with the reviewer's suggestion we have tempered our suggestions of causality throughout the manuscript and clarified that we are only reporting an association. Edits to text are marked in Track Changes. Specifically, we state in our limitations section that, *"Furthermore, while the differences we observe between cases and controls precede the onset of ASD symptoms, these data do not establish causality"*. Line 271.

In summary, this is an intriguing paper using novel methodology to suggest a possible etiological role for abnormalities in trace metals in autism. The manuscript has a number of strengths, but also limitations and I have focused on the later in this review. Nevertheless, I think it is of sufficient importance to be given serious consideration.

Authors' response: We thank the reviewer for their positive feedback and the very useful critique.

Reviewer #4 (Remarks to the Author):

I was asked to review this paper due to my expertise in laser ablation inductively coupled mass spectrometry (LA-ICP-MS). As such I will be limiting my comments to that aspect of the paper and to the best of my knowledge and ability will try to address the impact of the LA-ICP-MS on the conclusions. The biggest issue I have is the almost complete lack of experimental detail. It consists of a very cursory description of the equipment used with no information at all about laser parameters such as spot size, fluence, repetition rate or any similar information about the ICP-MS. Two references are given to earlier work but I only had access to one, Austin C, et al. Barium distributions in teeth reveal early-life dietary transitions in primates. Nature 498,

216-219 (2013). This reference contained slightly more detail about the laser, but it was a completely different manufacturer and type, New Wave, Nd:Yag at 213 nm vs ESI, ArF at 193 nm (the wavelength was not given, it is my own knowledge that filled that in. A different type of ICP-MS was used as well. As a result this reference gives me little to go on. There was yet another reference given here, but in a dental journal that I do not have access to.

In spite of this lack of detail, the data shown in figure 2 do seem to support the stated result of observed differences in the temporal accumulation of Pb, Mn and Zn between discordant pairs of twins and the relative similarities among concordant pairs. I would recommend this paper for publication with the addition of more detail on the LA-ICP-MS method as described above.

Authors' response: We apologize for not providing sufficient detail of our LA-ICP-MS methods. We have now added to the Supplementary Materials all the details of the laser and ICP-MS parameters used during operation, including the items mentioned by the reviewer (spot size, fluence, etc). Additionally, we have provided details of the manufacturer and set up (e.g. the use of a large cell format, types of flow gas used, etc.). Please below a copy of the section we have inserted in our Supplemental Material.

We would like to reassure the reviewers that we have a specialist laser ablation PhD chemist on the team (Dr. Austin) and have over one dozen publications on this analytical method.

We thank Reviewer 4 for recognizing the value of this work and supporting its publication.

Supplementary Method 3: Laser Ablation-Inductively Coupled Plasma-Mass Spectrometry

A New Wave Research NWR-193 (ESI, USA) laser ablation unit equipped with a 193nm ArF excimer laser was connected to an Agilent Technologies 8800 triple-quad ICP-MS (Agilent Technologies, USA). Helium was used as carrier gas from the laser ablation cell and mixed with argon via Y-piece before introduction to the ICP-MS. The system was tuned daily using NIST SRM 612 (trace elements in glass) to monitor sensitivity (maximum analyte ion counts), oxide formation ($^{232}\text{Th}^{16}\text{O}^+ / ^{232}\text{Th}^+$, < 0.3%) and fractionation ($^{232}\text{Th}^+ / ^{238}\text{U}^+$, $100 \pm 5\%$). The laser was scanned in dentine parallel to the DEJ from the dentine horn tip towards the tooth cervix. A pre-ablation scan was run to remove any surface contamination. Data were analyzed as metal to calcium ratios (e.g. $^{208}\text{Pb} : ^{43}\text{Ca}$) to control for any variations in mineral content within a tooth and between samples. LA-ICP-MS operating parameters are given in Supplementary Table 1.

Supplementary Table 1. LA-ICP-MS operating conditions.

NWR-193 Laser Conditions		Agilent 8800 ICP-MS Conditions	
Wavelength (nm)	193	RF power (W)	1350
Helium carrier flow (L min ⁻¹)	0.8	Argon carrier flow (L min ⁻¹)	0.6
Fluence (J cm ⁻¹)	5.0	Plasma gas flow (L min ⁻¹)	15
Repetition rate (Hz)	10	Sample Depth (mm)	4.0
Spot size (μm)	35	Scan mode	Peak hopping
Scan speed (μm s ⁻¹)	35	Integration time (ms)	50 – 55

Reviewers' comments:

Reviewer #2 (Remarks to the Author):

This revision represents a very conscientious attempt to respond to the critiques, modify the statistical approach in an appropriately conservative manner, systematically acknowledge limitations, and add subjects TO THE EXTENT POSSIBLE. In this sense it constitutes an exhaustive and sophisticated attempt to analyze what is available. Given the fact, however, that the sample remains prone to severe ascertainment bias and there is not a replication set for this particular discovery dataset, I do not believe that the weight of the evidence warrants publication in a journal of the stature of Nature Communications, because this amounts to no more than a highly sophisticated case series, and one with enough limitations and alternate explanations that it would be more appropriately reported in a specialty journal related to autism until convincingly replicated in a larger sample, with a study design that is more appropriate to address the scientific questions.

Reviewer #3 (Remarks to the Author):

I have carefully examined the authors' response to my questions and I am satisfied that the alterations to the manuscript adequately address these issues. In terms of the concerns I have raised with my expertise, the changes are satisfactory for publication.

Reviewer #5 (Remarks to the Author):

I have carefully reviewed the statistical methods both in the text and in the supplement. I have the following concerns about the statistical methods in the manuscript.

1. The authors claimed that the data from tooth-matrix biomarkers are time-series data. The manuscript did not mention how many data points were collected in the "time-series". If there were sufficient data points in the time-series, the small sample size in the manuscript should not be an issue. In the analysis using distributed lag models, the typical example is to investigate the association of daily air pollution and daily mortality in a certain city, with a few years' observation. In this case, even though the sample size is $n=1$ (because there is only one time series), the model is perfectly valid. However, I did not know if the time-series in this manuscript have sufficient data point in each time series to make the distributed lag model valid.
2. On page 7, it was unclear how the "intra-twin correlation" was adjusted for. No details were given.
3. Most importantly, this manuscript was very exploratory by nature. The findings were not validated with any independent datasets by independent researchers. I was concerned about the validity and generalizability of the conclusions.

Reviewer #6 (Remarks to the Author):

I certainly recommend publication. Here's a brief review:

I have looked over Arora et al.'s MS on the use of trace element zoning in teeth from twins to assess

whether there are significant differences in pre- and post-natal metal concentrations for ASD discordant twins. I was asked principally to evaluate the robustness of the analytics, i.e. the laser-ablation ICP-MS analysis.

Let me just say at the outset that it is very hard to botch this kind of analysis. You can use pretty much any kind of laser, ICP-MS, carrier gas, flow rate, etc. and recover a meaningful zoning profile. You may not be able to quantify concentrations, and there might be differential smearing of compositional zoning depending on what system you use. But as long as consistent methods are used, relative changes to concentration will be reproducible and can be compared among paired teeth. Only if the teeth were vastly different sizes (obviously not), or different methods were used to analyze each tooth of a pair (obviously not), then I might worry about the comparability.

I confess I'm no statistician, but statistics shouldn't (normally) tell you anything that you don't already know. Yes, it quantifies your views in an important way, but for data like these, you should be able to just look at them and tell what's up. The reproducibility of zoning from concordant twins from Fig 3a and 3c is about what I would expect for the reproducibility of trace element zoning. That is, if I measured two parallel lines through a single tooth, I'd expect them to look like the two profiles presented in those two figure panels. The fact that such similar profiles were collected from two different teeth of concordant children, and the fact that paired teeth from ASD discordant children in Fig. 3b are so different from each other is compelling evidence for the conclusions of the study. Yes, the number of ASD discordant comparisons is small. But the data are about as unequivocal as I've ever seen.

That being said, I do have a few thoughts for small improvements:

First, I would like to know the direction of the laser lines. Looking at figure 2, were they collected "vertically" from the tooth cusp to the pulp cavity, parallel to the dentine-enamel interface, some other direction? If it's in the text, I couldn't find it. It would be easy to add a line to Fig. 2 showing this.

Second, is Fig. 3c ever explicitly referenced? A single sentence around line 137 would help make the point about how different the patterns are in Fig. 3b compared to 3c.

Third, it might be helpful to mention that trace element zoning in teeth is expected for a variety of reasons (e.g. natural variations in the metal content of an environment) and is observed not only in humans but in other non-primate mammals as well. I think the best reference for this is Kohn et al. (2013; J Archaeol. Sci.), which shows zoning in Sr, Ba, and Zn, collected by LA-ICP-MS. Yes, Austin et al. show Ba maps for monkeys (that reference shows up twice, BTW). But saying something about the generality to other mammals would provide broader context for thinking about tooth trace elements. And it might give readers ideas of how trace element studies might be applied to other organisms.

We thank the reviewers for their encouragement and constructive feedback that has helped us further improve our work. Detailed responses are provided below to each of the points raised by the reviewers.

Reviewer #3: *I have carefully examined the authors' response to my questions and I am satisfied that the alterations to the manuscript adequately address these issues. In terms of the concerns I have raised with my expertise, the changes are satisfactory for publication.*

Author response: We thank the reviewer for their positive feedback and for being supportive of the publication of this work.

Reviewer #2: *This revision represents a very conscientious attempt to respond to the critiques, modify the statistical approach in an appropriately conservative manner, systematically acknowledge limitations, and add subjects TO THE EXTENT POSSIBLE. In this sense it constitutes an exhaustive and sophisticated attempt to analyze what is available. Given the fact, however, that the sample remains prone to severe ascertainment bias and there is not a replication set for this particular discovery dataset, I do not believe that the weight of the evidence warrants publication in a journal of the stature of Nature Communications, because this amounts to no more than a highly sophisticated case series, and one with enough limitations and alternate explanations that it would be more appropriately reported in a specialty journal related to autism until convincingly replicated in a larger sample, with a study design that is more appropriate to address the scientific questions.*

Author response: We appreciate the many points raised by the reviewer and respond to each one below:

Point 1 [very conscientious attempt to respond to the critiques]: As the reviewer has noted, we have invested much effort to address all the reviewers' comments.

Point 2 [sample remains prone to severe ascertainment bias]: In the first round of review and now here again, the reviewer has raised the issue of statistical ascertainment bias questioning the "independence of serial measurements of metal accumulation over successive epochs" (statement from the first round of review). This is an important issue and we would like to reassure the reviewer that distributed lag models account for this. As Reviewer 5 (expert in statistical methods) has pointed out, these statistical methods are suitable for such data ("*perfectly valid*") and are also used for air pollution modeling where pollution levels on one day influence those on subsequent days. One other point we wish to stress is that most studies using distributed lag models only utilize 95% confidence intervals, but we have applied an additional correction to convey a more conservative interpretation of our results – the Holm Bonferroni correction.

Point 3 [no replication set for this particular discovery dataset]: The editor has advised that replication set is not necessary. We wish to stress to the reviewer that these samples are extremely rare and we have contacted every major autism cohort in the US that we are aware of. No discordant twins with teeth samples were available. Similar responses were received from collaborators in other studies in the UK and Australia. We, therefore, undertook convenience recruitment at the first author's institution (Mount Sinai Hospital, New York) to find one discordant twin pair and the results from that pair

are in agreement with our main study (reported in Supplemental Information). We are cognizant that this does not constitute a complete validation study and we have edited the text of the manuscript to convey this limitation.

Changes to manuscript text:

On Line 48 (abstract), we have added “independent replication and larger population-based studies examining twin and non-twin samples are needed to extend these findings”

Point 4 [...represents a very conscientious attempt to respond to the critiques, modify the statistical approach in an appropriately conservative manner, systematically acknowledge limitations.....vs.... I do not believe that the weight of the evidence warrants publication in a journal of the stature of Nature Communications...].

Author response: We respect the reviewer’s views, but emphasize that our sample of participants is not a case series but rather a non-random sample of a population based cohort, and the results are of broad significance to autism and other neurodevelopmental disorders. We have no response to the point about suitable journals, as that appears to be a matter for the editorial team.

Reviewer #5: *I have carefully reviewed the statistical methods both in the text and in the supplement. I have the following concerns about the statistical methods in the manuscript.*

1. The authors claimed that the data from tooth-matrix biomarkers are time-series data. The manuscript did not mention how many data points were collected in the “time-series”. If there were sufficient data points in the time-series, the small sample size in the manuscript should not be an issue. In the analysis using distributed lag models, the typical example is to investigate the association of daily air pollution and daily mortality in a certain city, with a few years’ observation. In this case, even though the sample size is $n=1$ (because there is only one time series), the model is perfectly valid. However, I did not know if the time-series in this manuscript have sufficient data point in each time series to make the distributed lag model valid.

Author response: Our analytical method generates on average 152 data points per tooth, each point representing a distinct developmental time. This results in a temporal resolution of approximately 2 to 4 days in our samples. The number of data points and fine temporal resolution allows us to use distributed lag models in a manner similar to air pollution studies. We have edited the methods text to indicate number of measures per sample. Line 327: On average, each tooth was sampled at 152 locations.

2. On page 7, it was unclear how the “intra-twin correlation” was adjusted for. No details were given.

Author response: We applied two additional corrections; first, we adjusted for intra-twin correlations through a random effect term per twin pair and, second, we used a Holm-Bonferroni correction to account for multiple comparisons (Line 370-372).

3. Most importantly, this manuscript was very exploratory by nature. The findings were

not validated with any independent datasets by independent researchers. I was concerned about the validity and generalizability of the conclusions.

Author response: This is the first study of this kind on an important and extremely rare sample of twins discordant for autism spectrum disorder. Although the sample size appears small, it actually represents a rather large minority of all potential cases. Still, we completely agree with the reviewer that replication in larger twin samples and non-twins case-control samples is important and have stated upfront in the Abstract that other populations need to be evaluated for the risk factors we have identified.

On Line 48 (abstract), we have added “independent replication and larger population-based studies examining twin and non-twin samples are needed to extend these findings”

Reviewer #6: *First, I would like to know the direction of the laser lines. Looking at figure 2, were they collected “vertically” from the tooth cusp to the pulp cavity, parallel to the dentine-enamel interface, some other direction? If it’s in the text, I couldn’t find it. It would be easy to add a line to Fig. 2 showing this.*

Second, is Fig. 3c ever explicitly referenced? A single sentence around line 137 would help make the point about how different the patterns are in Fig. 3b compared to 3c.

Third, it might be helpful to mention that trace element zoning in teeth is expected for a variety of reasons (e.g. natural variations in the metal content of an environment) and is observed not only in humans but in other non-primate mammals as well. I think the best reference for this is Kohn et al. (2013; J Archaeol. Sci.), which shows zoning in Sr, Ba, and Zn, collected by LA-ICP-MS. Yes, Austin et al. show Ba maps for monkeys (that reference shows up twice, BTW). But saying something about the generality to other mammals would provide broader context for thinking about tooth trace elements. And it might give readers ideas of how trace element studies might be applied to other organisms.

Author response: We thank the reviewer for their thoughtful comments and providing a very appropriate reference.

Point 1 [direction of laser lines]: This is an important point the reviewer has raised. The laser scan was conducted parallel to the dentine-enamel junction from the dentine horn to the cervix of the tooth crown.

As suggested by the reviewer, we have added a line to Figure 2 to indicate the location of sampling, and edited the figure legend accordingly.

Point 2 [referencing Figure 3c]. We thank the reviewer for their suggestion and have added the following statement and referenced Fig 3c.

Line 138-140: In MZ pairs concordant for ASD (Fig. 3c), the differences in metal distribution amongst twins were smaller than those observed in discordant pairs.

Point 3 [trace element zoning in teeth]. We have added the following sentences to the text to convey that metal zoning in teeth can occur due to multiple reasons.

Line 278: Heterogeneous distribution of metals in enamel and dentine can occur in response to variable environmental exposures, changes in diet and due to age-

dependent metabolic changes, and has been observed in humans and other mammals.³⁸

Reference 38. Kohn MJ, Morris J, Olin P. Trace element concentrations in teeth – a modern Idaho baseline with implications for archeometry, forensics, and palaeontology. *J Archaeol Sci.* **40**, 1689-1699 (2013).

Duplicated reference to Austin et al. has been corrected. We thank the reviewer for pointing this out.

REVIEWERS' COMMENTS:

Reviewer #5 (Remarks to the Author):

As I pointed out in my prior review, I had major concerns about exploratory nature of the manuscript and there was no independent datasets to validate the findings. The authors did add some comments about this fact. Whether this is acceptable will be decided by the editor.

Reviewer #6 (Remarks to the Author):

The authors have adequately addressed my concerns. Please publish as is.

Response to reviewer comments

Reviewer #5 (Remarks to the Author):

As I pointed out in my prior review, I had major concerns about exploratory nature of the manuscript and there was no independent datasets to validate the findings. The authors did add some comments about this fact. Whether this is acceptable will be decided by the editor.

Reviewer #6 (Remarks to the Author):

The authors have adequately addressed my concerns. Please publish as is.

We thank the reviewers for their time in considering our edits, and appreciate their support for the publication of our manuscript. No changes have been suggested and therefore we have not made any further edits to the manuscript.